# PRIVACY-AWARE DATA INTEGRATION FOR ENHANCED QUANTILE INFERENCE UNDER HETEROGENEITY

## ABSTRACT

Quantile estimation and inference play essential roles in diverse scientific and industrial applications, and their accuracy can often be enhanced by integrating auxiliary data from multiple sites. However, developing efficient aggregation methods for quantile inference under potential privacy constraints, particularly with heterogeneous datasets, remains challenging. To address these issues, we propose a systematic framework for quantile estimation and inference under potential local differential privacy (LDP). The key idea is to construct weighted estimators by adaptively aggregating quantile estimates from target and source sites. The adaptive weights are determined by minimizing the asymptotic variance, incorporating an additional $\ell_2$ penalty to account for parameter shift. A parallel stochastic gradient descent algorithm under LDP constraints is developed for weight estimation and valid inference. Additionally, we introduce a conservative weighted estimator to ensure robust inference across diverse heterogeneous scenarios. Rigorous theoretical analysis establishes the consistency, normality, and effectiveness of the proposed methods. Extensive numerical studies and real data application corroborate our theoretical findings.

## 1 INTRODUCTION

**Motivation.** Quantile estimation and inference play a central role in many scientific and industrial applications (Chernozhukov & Fernández-Val, 2011; Huang et al., 2017; Kallus et al., 2024; Deuber et al., 2024; Yadlowsky et al., 2025). Quantiles provide robust summaries of outcome distributions, especially in the presence of heavy tails or extreme events. For example, financial institutions routinely use quantile-based risk measures such as value-at-risk to quantify and manage investment risk (Chen, 2008; Barbaglia et al., 2023). Because such quantities directly inform risk management and operational decisions, improving the accuracy and efficiency of quantile inference has attracted substantial attention.

A natural way to enhance quantile estimation is to leverage auxiliary datasets collected by multiple organizations or sites, thereby integrating information beyond a single target dataset (Wang et al., 2019; Cai et al., 2024a; Han et al., 2025). However, many real-world datasets contain sensitive personal or proprietary information and are subject to ethical and legal protections (Dwork et al., 2006; Cai et al., 2024c). In practice, privacy requirements often differ across sites: hospital networks, financial consortia, and federated platforms (e.g., smartphones or autonomous vehicles) operate under heterogeneous jurisdictional and organizational rules. As a result, some sites may release statistics that satisfy record-level local differential privacy (LDP), while others may only share unperturbed summaries or aggregate statistics (Konečný et al., 2016; Hard et al., 2018; Li et al., 2020; Nguyen et al., 2022). This raises a fundamental question:

> How can we effectively integrate auxiliary data sources to improve quantile estimation and inference, while respecting site-specific privacy constraints through mechanisms such as differential privacy?

**Challenges.** A growing literature studies data integration and federated inference; see Section 2 for a detailed review. Nonetheless, existing approaches face several obstacles in the setting of privacy-aware quantile inference. First, many integration methods construct weighted estimators that com-

bine estimators from the target and auxiliary datasets, choosing the weights by minimizing criteria related to asymptotic variance (Li et al., 2022a; Cai et al., 2024a). In quantile problems under LDP, however, commonly used variance estimation tools such as classical sample variance or plug-in estimators (Zhu et al., 2021; Li et al., 2023; Huang et al., 2022; Gu & Chen, 2023; Han et al., 2025; Guo et al., 2025) are no longer directly applicable, because the locally privatized data substantially alter the underlying data distribution. Even without privacy constraints, most existing data integration methods focus on mean-type parameters or the minimization of smooth loss functions (Li et al., 2013; Lee et al., 2017; Chen & Xie, 2014; Li et al., 2022a; 2023; Zhu et al., 2024), whereas quantile loss functions are non-smooth and violate the regularity assumptions those methods rely on.

Second, current integration strategies typically impose restrictive assumptions on the relationship between target and auxiliary data sources. Many works assume that source and target parameters are exactly identical (Lee et al., 2017; Duan et al., 2020; Zhu et al., 2021; Wang & Shen, 2024), or that their differences are separated by a margin bounded away from zero (Huang et al., 2022; Li et al., 2022a; Cai et al., 2024b;c). Such assumptions can be unrealistic in heterogeneous multi-site environments and do not guarantee valid inference across the diverse scenarios encountered in practice.

**Contributions.** To address these challenges, we develop a systematic framework for improving quantile estimation and inference when both the target and auxiliary source datasets may be subject to LDP constraints.

- Our main idea is to construct weighted quantile estimators that adaptively combine information from the target and source sites. The weights are chosen to minimize the asymptotic variance of the resulting estimator, with an additional $\ell_2$ penalty that regularizes parameter shifts between target and source sites. To implement this strategy under LDP, we design a parallel stochastic gradient descent (PSGD) algorithm that estimates the optimal weights while respecting local privacy constraints and enabling valid downstream inference. We further introduce a conservative weighted estimator that yields robust inference across a broad range of heterogeneous scenarios.

- Methodologically, our work provides a general and flexible framework for privacy-aware quantile estimation and inference via data integration. The proposed family of weighted estimators can substantially improve both estimation accuracy and inferential validity for the target quantile parameter under suitable conditions, and the framework accommodates diverse forms of heterogeneity across sites.

- Theoretically, we establish rigorous guarantees for the proposed procedures with the typical non-smooth quantile loss function: (i) we prove consistency of the variance estimator obtained from the PSGD algorithm, providing a sound basis for the weighted estimators and their associated inference; (ii) we derive consistency and asymptotic normality of the resulting weighted quantile estimators under heterogeneous multi-site settings; and (iii) we show that, under mild conditions and whenever auxiliary sites contain useful information, our methods achieve strictly better estimation and inference performance than using the target site alone.

The rest of the paper is organized as follows. Section 2 provides a brief literature review on data integration and LDP. Section 3.1 introduces the methodologies and complete algorithms. Theoretical results are presented in Section 4, with corresponding assumptions and discussions in Section A. Extensive numerical results and real data analysis are given in Section 5, and additional simulations are listed in Section B. All technical lemmas and proofs can be found in Sections C and D.

## 2 RELATED WORK

**Data integration.** Recently, statistical data integration methods have attracted growing interest. Under the assumption of parameter homogeneity, existing studies primarily develop aggregation strategies that minimize appropriately defined asymptotic variance criteria of parameter estimators (Li et al., 2013; Chen & Xie, 2014; Wang et al., 2019; Zhu et al., 2021; Gu & Chen, 2023). When potential parameter shift exists, aggregation strategies in the existing literature typically also consider biases between parameters from auxiliary sources and the target parameter to mitigate adverse effects (Li et al., 2022a; 2023; Cai et al., 2024a;c;b; Han et al., 2025). To mitigate privacy concerns,

classical methods aggregate summarized statistics (e.g., parameter estimates) rather than raw data (Chen et al., 2006; Lee et al., 2017; Duan et al., 2020; Guo et al., 2025; Bai et al., 2024), while recent approaches incorporate differential privacy constraints to achieve stronger privacy guarantees (Cai et al., 2024a;c;b). In particular, statistical data integration methods have also been extensively developed for quantile problems. The existing literature mainly focuses on estimation (Hu et al., 2021; Jiang & Yu, 2021; Tan et al., 2022; Pillutla et al., 2024; Wang & Shen, 2024; Shi et al., 2025). Recently, a few studies have also addressed inference problems (Huang et al., 2022; Bai et al., 2024).

**Local differential privacy (LDP).** Differential privacy (DP) bounds how much a statistic can change if one record is modified, formalizing "plausible deniability" Dwork et al. (2006). Variants such as Rényi DP, zCDP, and concentrated DP sharpen composition and enabled releases like the 2020 U.S. Census. DP's Achilles' heel is its reliance on a trusted curator; breaches, subpoenas, or misconfigurations can expose raw data Narayanan & Shmatikov (2008). Local DP (LDP) removes that trust by randomizing data at the source, generalizing randomized response Kasiviswanathan et al. (2011); Duchi et al. (2013). Pan-DP further shows only locally perturbed data withstand repeated intrusions, aligning pan-DP with LDP Amin et al. (2020). LDP is now deployed in Chrome telemetry, Safari domain statistics, and Windows Defender reporting. These applications demonstrate that curator-free privacy can coexist with high-utility analytics, spurring research on utility-optimal protocols, adaptive privacy budgeting, and federated inference.

## 3 METHODOLOGY

### 3.1 PROBLEM DESCRIPTION

We begin by introducing the model setup and notation. Due to page limitations, a complete list of notation is provided in Appendix C. Consider a total of $N$ observations stored across a fixed set of $K + 1$ sites, indexed by $\{0, 1, \ldots, K\}$. Denote the sample size at the $k$-th site by $n_k$, with $\sum_{k=0}^{K} n_k = N$, and assume that $n_k \asymp N/K$. At each site $k$, observations $\{X_{k,t}\}_{t=1}^{n_k} \subseteq \mathbb{R}$ are independently generated from an unknown distribution $\mathcal{P}_k$. The parameter of interest at each site is the quantile at a specified level $\tau \in (0, 1)$. Specifically, define the check loss function as: $\ell(x, \theta) = (x - \theta)(\tau - \mathbf{1}(x \leq \theta))$. Then, the quantile parameter at the $k$-th site satisfies that:

$$\theta_k = \arg\min_{\theta} \mathbb{E}_{x \sim \mathcal{P}_k}\{\ell(x, \theta)\}. \tag{3.1}$$

To estimate the parameter $\theta_k$ in practice, one typically minimizes the empirical counterpart of the objective function, which is given by $\widehat{\theta}_k = \arg\min_{\theta} \sum_{t=1}^{n_k} \ell(X_{k,t}, \theta)$. Under regular conditions, it is assumed that $\widehat{\theta}_k$ admits the following asymptotic rule:

$$\sqrt{n_k}\left(\widehat{\theta}_k - \theta_k\right) \xrightarrow{d} \mathcal{N}\left(0, \{\tau(1-\tau)\}/\{f_k^2(\theta_k)\}\right), \tag{3.2}$$

where $f_k$ denotes the probability density function associated with the distribution $\mathcal{P}_k$. Various algorithms are available for solving this empirical optimization problem, facilitating both estimation and statistical inference for $\theta_k$. The classical and simplest method is based on order statistics (Van der Vaart, 2000). Specifically, the quantile parameter $\theta_k$ at site $k$ can be directly estimated by taking the corresponding empirical quantile. Inference is typically conducted by plugging in a density estimator, such as a kernel density estimator, for the unknown probability density function evaluated at $\tau$. While this method is straightforward and efficient, it directly utilizes raw data, limiting its applicability in sensitive scenarios. An alternative classical approach is Averaged Stochastic Gradient Descent (ASGD) (Polyak & Juditsky, 1992; Chen et al., 2023). Starting from an initial estimator $\widehat{\theta}_{k,0}$, ASGD iteratively updates the estimator at each site $k$ as follows:

$$\widehat{\theta}_{k,t+1} = \widehat{\theta}_{k,t} - \eta_{k,t}\left\{\tau - \mathbf{1}\left(X_{k,t+1} \leq \widehat{\theta}_{k,t}\right)\right\}, \tag{3.3}$$

where $0 \leq \eta_{k,t} \leq 1$ denotes the learning rate. The final estimator is computed as the average of all iterates as $\widehat{\theta}_k = n_k^{-1} \sum_{t=1}^{n_k} \widehat{\theta}_{k,t}$. Statistical inference can be conveniently implemented using self-normalized methods (Li et al., 2022b; Lee et al., 2022). Compared to the order-statistics-based method, the iterative nature of ASGD makes it naturally amenable to incorporating privacy-preserving mechanisms. Specifically, we consider here the ASGD algorithm under LDP constraints (Liu et al., 2023). Before introducing the detailed algorithm, we first provide formal definitions for DP and LDP.

**Definition 1** (DP, see (Dwork et al., 2006)). *A randomized algorithm $\mathcal{A}$, taking a dataset consisting of individuals as its input, is $(\epsilon, \delta)$-differentially private if, for any pair of datasets $S$ and $S'$ that differ in the record of a single individual and any event $E$, satisfies $\mathbb{P}[\mathcal{A}(S) \in E] \leq e^{\epsilon}\mathbb{P}[\mathcal{A}(S') \in E] + \delta$. When $\delta = 0$, $\mathcal{A}$ is called $\epsilon$-differentially private ($\epsilon$-DP).*

**Definition 2** (LDP, see Joseph et al. (2019)). *An $(\epsilon, \delta)$-randomizer $R : X \to Y$ satisfies $(\epsilon, \delta)$-LDP if, for any event $E$ and any input data point $X \neq X'$, $\mathbb{P}[R(X) \in E] \leq e^{\epsilon}\mathbb{P}[R(X') \in E] + \delta$.*

Next, we modify the classical ASGD procedure in (3.3) by incorporating a local randomization step into the binary indicator function $\mathbf{1}(X_{k,t+1} \leq \widehat{\theta}_{k,t})$. To be more precise, at the $t$-th iteration, we issue a query to the private data point $X_{k,t+1}$. In response, with probability $r_k$, we receive the true binary indicator, and with probability $1 - r_k$, we receive an random variable $v \sim \text{Bernoulli}(0.5)$; see detailed Algorithm B.1 in Appendix. Since the Algorithm B.1 enjoys $(\epsilon_k, 0)$-LDP with $\epsilon_k = \log\{(1 + r_k)/(1 - r_k)\}$, our Algorithms 1 and B.2 have $(\max_{1 \leq k \leq K} \epsilon_k, 0)$-LDP guarantees as a direct consequence. Here, the response rate $r_k$ controls the level of privacy protection, with smaller values corresponding to stronger privacy guarantees. When $r_k = 1$, the method reduces to the standard non-private case. Since the observed binary variable is now a randomized version of the original indicator, it is necessary to execute bias-correction to ensure an unbiased gradient estimate. Let $\widehat{\zeta}_{k,t}$ denote the perturbed binary variable observed at iteration $t$ and site $k$. Under this LDP mechanism, the iterative updating formula in (3.3) becomes:

$$\widehat{\theta}_{k,t+1} = \widehat{\theta}_{k,t} - \eta_{k,t}\left\{\frac{1 + r_k - 2r_k\tau}{2}\widehat{\zeta}_{k,t} - \frac{1 - r_k + 2r_k\tau}{2}\left(1 - \widehat{\zeta}_{k,t}\right)\right\}. \tag{3.4}$$

Similarly as in the classical ASGD method, the final estimator is obtained by averaging the iterates over $n_k$ steps. Statistical inference can then be performed using self-normalization techniques adapted to this LDP setting, see (Liu et al., 2023).

The methods described above provide feasible algorithms for solving the optimization problem (3.1) at each site $k$, with possible LDP constraints. Specifically, the classical order-statistics approach can be applied directly if privacy protection is unnecessary, whereas the ASGD-based method should be employed in an LDP setting. However, the parameter estimation and inference at each site can potentially be improved further by appropriately aggregating data across multiple sites, especially if other sites contain useful and relevant information, such as sharing the same underlying quantile parameter. Without loss of generality, we treat site 0 as the target site and the remaining $K$ sites as source sites.

## 3.2 Adaptive Weighted Estimator

A natural way to utilize information from multiple sites is to combine the estimators from the target and source sites into a weighted estimator. Specifically, for each site $k$, let $\widehat{\theta}_k$ denote the estimator of $\theta_k$ derived by one of the previously discussed methods, and let $w_k$ be the corresponding weight satisfying $w_k \geq 0$ for all $0 \leq k \leq K$ and $\sum_{k=0}^{K} w_k = 1$. We then define the weighted estimator $\widehat{\theta}(\mathbf{w}) = \sum_{k=0}^{K} w_k \widehat{\theta}_k$, where $\mathbf{w} = \{w_k\}_{k=0}^{K}$. Our goal is to determine weights $\{w_k\}$ that maximize the efficiency of the weighted estimator while controlling for the negative impact arising from heterogeneity in data distributions and parameters between the target and source sites. To this end, we introduce the following loss function with respect to the weights $\mathbf{w}$:

$$\mathcal{L}(\mathbf{w}) = \sum_{k=0}^{K} w_k^2 \sigma_k^2 / n_k + \lambda \sum_{k=0}^{K} w_k^2 b_k^2,$$

where $\sigma_k^2 / n_k$ is the asymptotic variance of the estimator $\widehat{\theta}_k$. We will rigorously prove that $\sigma_k^2 = \left\{4r_k^2 f_k^2(\theta_k)\right\}^{-1}\left\{1 - r_k^2(2\tau - 1)^2\right\}$ in subsequent theoretical analysis. The bias term $b_k = \theta_k - \theta_0$ represents the parameter shift of the $k$-th source site relative to the target site 0. The tuning parameter $\lambda \geq 0$ controls the trade-off between variance and bias. In particular, setting $\lambda = 0$ yields classical inverse-variance weighting (Zhu et al., 2021; Shi et al., 2023), while setting $\lambda = 1$ approximately corresponds to minimizing the mean squared error of the estimator (Li et al., 2023). Minimizing $\mathcal{L}(\mathbf{w})$ with respect to $\mathbf{w}$ yields the oracle weights $\mathbf{w}^* = \arg\min_{\mathbf{w}} \mathcal{L}(\mathbf{w})$, which has the following

explicit closed-form solution:

$$w_k^* = \left\{ \sum_{j=0}^{K} \left( \frac{\sigma_j^2}{n_j} + \lambda b_j^2 \right)^{-1} \right\}^{-1} \left( \frac{\sigma_k^2}{n_k} + \lambda b_k^2 \right)^{-1}, \quad 0 \le k \le K. \tag{3.5}$$

Here, oracle weights refer to the ideal weights that minimize the estimator's asymptotic variance under the assumption that the true site-specific variances are known. The weights in (3.5) are adaptive in that they automatically adjust to each source site's characteristics: sites with higher noise (e.g., lower response rates $r_k$) or larger parameter shifts (biases $b_k$) receive smaller weights, while more stable and better-aligned sites receive larger weights. This allows the weighted estimator to prioritize sources that are most informative for the target site.

In practice, the oracle weights in (3.5) involve unknown quantities and must be estimated from data. A natural estimator for the bias term is $\widehat{b}_k = \widehat{\theta}_k - \widehat{\theta}_0$. For the variance term $\sigma_k^2$, the classical plug-in estimator is often used when an explicit form is available (Han et al., 2025). However, in our setting some sites operate under LDP constraints, and the plug-in method becomes infeasible because the raw data needed to estimate $f_k$ cannot be accessed. To overcome this limitation, we propose a PSGD approach that estimates $\sigma_k^2$ automatically when raw data are unavailable. The key idea is to partition each site's local data into multiple subsets and run independent SGD procedures—referred to as chains—on these subsets, enabling direct variance estimation.

Specifically, at each site $k$, the data $\{X_{k,t}\}_{t=1}^{n_k}$ are randomly divided into $M_k$ subsets, each forming an i.i.d. SGD chain. Let $\{X_{k,t}^{(m)}\}_{t=1}^{\lfloor n_k/M_k \rfloor}$ denote the data in chain $m$ for $1 \le m \le M_k$. For each chain, PSGD initializes the estimator $\widehat{\theta}_{k,0}^{(m)} = \widehat{\theta}_{k,0}$ for $0 \le k \le K$ and updates it iteratively as follows:

$$\widehat{\theta}_{k,t+1}^{(m)} = \widehat{\theta}_{k,t}^{(m)} - \eta_{k,t} \left\{ \frac{1 + r_k - 2r_k\tau}{2} \widehat{\zeta}_{k,t}^{(m)} - \frac{1 - r_k + 2r_k\tau}{2} \left( 1 - \widehat{\zeta}_{k,t}^{(m)} \right) \right\}, \tag{3.6}$$

where $\widehat{\zeta}_{k,t}^{(m)}$ denotes the locally randomized version of the indicator $\mathbf{1}(X_{k,t+1}^{(m)} \le \widehat{\theta}_{k,t}^{(m)})$. After completing the iterations within each chain, we compute the chain-specific estimator by $\widehat{\theta}_k^{(m)} = (\lfloor n_k/M_k \rfloor)^{-1} \sum_{t=1}^{\lfloor n_k/M_k \rfloor} \widehat{\theta}_{k,t}^{(m)}$. The final estimator at site $k$ is then obtained by averaging across the $M_k$ chain-specific estimators: $\widehat{\theta}_k = M_k^{-1} \sum_{m=1}^{M_k} \widehat{\theta}_k^{(m)}$. Note that these chain-specific estimators $\{\widehat{\theta}_k^{(m)}\}$ are independent of each other. This inspires the following variance estimator:

$$\widehat{\sigma}_k^2 = (M_k - 1)^{-1} \sum_{m=1}^{M_k} \lfloor n_k/M_k \rfloor \left( \widehat{\theta}_k^{(m)} - \widehat{\theta}_k \right)^2. \tag{3.7}$$

Next, we estimate the oracle weights defined in equation (3.5) by replacing the unknown parameters $\sigma_k^2$ and $b_k$ with their estimators $\widehat{\sigma}_k^2$ and $\widehat{b}_k$, respectively. We denote these estimated weights as $\{\widehat{w}_k\}$. Subsequently, the resulting weighted estimator for the target parameter $\theta_0$ and corresponding variance estimator are obtained as

$$\widehat{\theta}_{\text{est}} = \sum_{k=0}^{K} \widehat{w}_k \widehat{\theta}_k, \quad \widehat{\sigma}_{\text{est}}^2 = \sum_{k=0}^{K} \frac{N}{n_k} \widehat{w}_k \widehat{\sigma}_k^2.$$

We will theoretically show that $\widehat{\sigma}_{\text{est}}^2/N$ is a consistent estimator of $\text{Var}(\widehat{\theta}_{\text{est}})$. Thus, statistical inference for $\widehat{\theta}_{\text{est}}$ can be readily conducted by constructing a $(1 - \alpha)$-confidence interval: $[\widehat{\theta}_{\text{est}} - \mathcal{Z}_{\alpha/2}\widehat{\sigma}_{\text{est}}/\sqrt{N}, \widehat{\theta}_{\text{est}} + \mathcal{Z}_{\alpha/2}\widehat{\sigma}_{\text{est}}/\sqrt{N}]$, where $\mathcal{Z}_{\alpha/2}$ is the upper $(\alpha/2)$-quantile of the standard normal distribution. The complete pseudo code for the data integration algorithm is described below in Algorithm 1.

### 3.3 CONSERVATIVE WEIGHTED ESTIMATOR

Algorithm 1 provides an adaptive weighting scheme capable of automatic inference under possible LDP constraints. Its validity, however, relies on conditions governing the parameter shifts between

---

**Algorithm 1** Privacy-Aware Quantile Inference via Data Integration

---

**Input:** Learning rates $\{\eta_{k,t}\}$, sample sizes $\{n_k\}$, $N = \sum_k n_k$, number of chains $\{M_k\}$, target quantile $\tau$, truthful response rates $\{r_k\}$, tuning parameter $\lambda$, and significance level $\alpha$.

**Output:** $\widehat{\theta}_{\mathrm{est}}$ and $[\widehat{\theta}_{\mathrm{est}} - \mathcal{Z}_{\alpha/2}\widehat{\sigma}_{\mathrm{est}}/\sqrt{N}, \widehat{\theta}_{\mathrm{est}} + \mathcal{Z}_{\alpha/2}\widehat{\sigma}_{\mathrm{est}}/\sqrt{N}]$.

**Initialization:** set $\widehat{\theta}_{k,0}^{(m)} \leftarrow 0$ for all $k, m$.

 **for** $0 \leq k \leq K$ **do**

  **for** $1 \leq m \leq M_k$ **do**

   **for** $1 \leq t \leq \lfloor n_k/M_k \rfloor$ **do**

    Obtain the locally randomizer $\widehat{\zeta}_{k,t}^{(m)} = \mathrm{LRC}\left(\widehat{\theta}_{k,t}^{(m)}, r_k, X_{k,t+1}^{(m)}\right)$ using Algorithm B.1.

    Compute $\widehat{\theta}_{k,t}^{(m)}$ according to equation (3.6).

   **end for**

  Compute the chain-specific estimator by $\widehat{\theta}_k^{(m)} = (\lfloor n_k/M_k \rfloor)^{-1} \sum_{t=1}^{\lfloor n_k/M_k \rfloor} \widehat{\theta}_{k,t}^{(m)}$.

  **end for**

 Compute the final estimator and corresponding variance estimator at site $k$ by

 $\widehat{\theta}_k = M_k^{-1} \sum_{m=1}^{M_k} \widehat{\theta}_k^{(m)}, \quad \widehat{\sigma}_k^2 = (M_k - 1)^{-1} \sum_{m=1}^{M_k} \lfloor n_k/M_k \rfloor (\widehat{\theta}_k^{(m)} - \widehat{\theta}_k)^2$.

 **end for**

 Compute the weighted estimator and its variance estimator by $\widehat{\theta}_{\mathrm{est}} = \sum_{k=0}^K \widehat{w}_k \widehat{\theta}_k$ and

 $\widehat{\sigma}_{\mathrm{est}}^2 = \sum_{k=0}^K N \widehat{w}_k \widehat{\sigma}_k^2 / n_k$, where $\widehat{w}_k = \left\{ \sum_{j=0}^K \left( \frac{\widehat{\sigma}_j^2}{n_j} + \lambda \widehat{b}_j^2 \right)^{-1} \right\}^{-1} \left( \frac{\widehat{\sigma}_k^2}{n_k} + \lambda \widehat{b}_k^2 \right)^{-1}$.

---

target and source sites. In particular, the bias $b_k$ must be either negligible or clearly distinguishable, that is, much smaller or much larger than $N^{-1/2}$. This requirement is weaker than most assumptions in prior work (Shi et al., 2023; Gu & Chen, 2023; Han et al., 2025), but it omits an important intermediate regime where $b_k$ is exactly of order $N^{-1/2}$.

In this regime, even oracle weights cannot ensure valid inference. The intuition is straightforward. Consider the case with a single source site ($K = 1$). When $b_1 \ll N^{-1/2}$, the resulting bias is dominated by the estimator's variance and can be ignored. When $b_1 \gg N^{-1/2}$, the oracle assigns almost no weight to site 1, mitigating its impact. But when $b_1 \asymp N^{-1/2}$, the bias is of the same order as the standard errors of $\widehat{\theta}_0$ and $\widehat{\theta}_1$. In this setting, the oracle cannot down-weight site 1 sufficiently, leaving the final estimator with a bias comparable to its variance and invalidating inference. Our proposed estimator necessarily inherits this limitation. Figure 1 depicts the contrast between cases where the bias is negligible and where it competes with the variance.

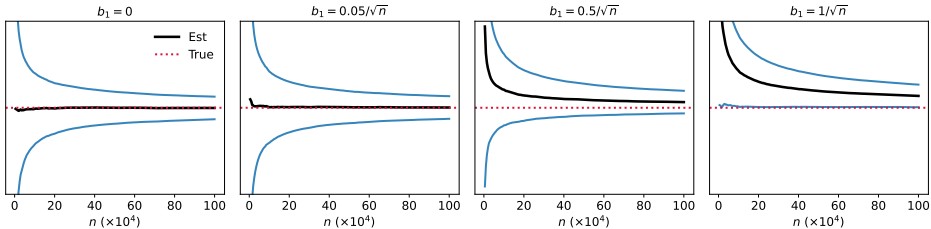

Figure 1: Illustration of confidence intervals under varying levels of bias. Each panel shows an estimator (black line) with bias $b_1$ and known variance $1/n$. The corresponding confidence interval with 95% significance level is represented by the blue lines. The red dashed line represents the true parameter value. The bias increases gradually from left to right. In the rightmost panel, the confidence interval no longer covers the true parameter, indicating that inference becomes invalid.

To address this issue, we propose a conservative approach based on the adaptive weighting method in Section 3.2. The key idea is to construct a conservative estimate of $b_k$, which tends to further down-weight biased source sites. Specifically, we replace the direct bias estimates with conservative upper bounds as

$$\widetilde{b}_k = |\widehat{b}_k| + \mathcal{C}\sqrt{\widehat{\sigma}_k^2/n_k + \widehat{\sigma}_0^2/n_0}, \ \ \mathcal{C} > 0,$$

where $\widehat{\sigma}_k^2/n_k + \widehat{\sigma}_0^2/n_0$ is the asymptotic variance of $\widehat{b}_k$, and $\mathcal{C}$ controls the conservativeness level. We then derive the conservative weights $\widetilde{w}_k$ for $0 \le k \le K$ by replacing $\sigma_k^2$ and $b_k$ in equation (3.5) with $\widehat{\sigma}_k^2$ and $\widetilde{b}_k$, respectively, and setting $\widetilde{b}_0 = 0$. Accordingly, the conservative weighted estimator for the target parameter $\theta_0$ is defined as: $\widehat{\theta}_{\mathrm{cons}} = \sum_{k=0}^K \widetilde{w}_k \widehat{\theta}_k$. The corresponding variance estimator for $\widehat{\theta}_{\mathrm{cons}}$ is given by $\widehat{\sigma}_{\mathrm{cons}}^2 = \sum_{k=0}^K N\widetilde{w}_k \widehat{\sigma}_k^2/n_k$. A complete algorithm is given in B.2.

## 4 THEORETICAL PROPERTIES

We first establish the privacy and statistical guarantee for the proposed PSGD algorithm. Due to the space limit, the complete assumptions and corresponding comments are provided in Appendix A. Let $\{\widehat{\theta}_k\}$ and $\{\widehat{\sigma}_k^2\}$ be the estimators produced by the PSGD in Algorithm 1 (B.2). It is worth noting that the entire algorithm to solve for $\widehat{\theta}_k, \widehat{\sigma}_k^2$ We first provide the LDP guarantee for the PSGD algorithm, which is summarized in Proposition 4.1. Subsequently, the asymptotic properties of these estimators are established in Theorem 4.1.

**Proposition 4.1** (Differential privacy). *The PSGD algorithm for the $k$-th site satisfies $(\epsilon_k, 0)$-LDP with $\epsilon_k = \log\{(1+r_k)/(1-r_k)\}$. Therefore, both the Algorithms 1 and B.2 integrating $K+1$ sites LDP data are $(\max_{0 \le k \le K} \epsilon_k, 0)$- LDP.*

**Theorem 4.1** (PSGD algorithm). *Under Assumptions A.1 – A.3, for each $k = 0, \ldots, K$, we have $\widehat{\sigma}_k^2 - \left\{4r_k^2 f_k^2(\theta_k)\right\}^{-1}\left\{1 - r_k^2(2\tau - 1)^2\right\} = \mathcal{O}_p(1)$ and $\widehat{\theta}_k$ satisfies*

$$\sqrt{n_k}\left(\widehat{\theta}_k - \theta_k\right) \xrightarrow{d} \mathcal{N}\left(0, \frac{1 - r_k^2(2\tau - 1)^2}{4r_k^2 f_k^2(\theta_k)}\right).$$

Theorem 4.1 extends the classical PSGD framework (Zhu et al., 2024) to the non-smooth quantile loss, delivers consistent variance estimation, and establishes the asymptotic normality of the estimator at the $k$-th site. We find that larger values of $r_k$ lead to smaller asymptotic variance, but they also correspond to weaker privacy guarantees.

Next, we provide theoretical guarantees for the weighted estimators. In our theoretical analysis, we investigate the properties of the proposed estimators under various choices of $\lambda$. Specifically, under the assumptions stated below, Theorem 4.2 establishes the validity of statistical inference for these weighted estimators and efficiency improvement.

**Assumption 1** (The choice of $\lambda$). *Assume that $\lambda = \mathcal{O}(1)$ and $\lambda b_k \gg N^{-1/2}$ for every $b_k \gg N^{-1/2}$.*

**Assumption 2** (Bias scale). *Assume the bias $b_k$ for each $1 \le k \le K$ satisfies at least one of the following conditions: (1) Vanishing bias: $b_k \ll N^{-1/2}$ or (2) Distinguishable bias: $b_k \gg N^{-1/2}$.*

**Theorem 4.2** (The property of $\widehat{\theta}_{\mathrm{est}}$). *Under Assumption A.4, assume that one of the following three scenarios holds:*

*(a) $\lambda = 0$, the bias scale satisfies Assumption 2 (1).*

*(b) $\lambda$ is bounded away from 0, the bias scale satisfies Assumption 2 (2).*

*(c) $\lambda$ satisfies Assumption 1, the bias scale satisfies Assumption 2.*

*Then we have (i) $\widehat{w}_k - w_k^* = \mathcal{O}_p(1)$ for $1 \le k \le K$, (ii) $\widehat{\sigma}_{\mathrm{est}}^2 - \sum_{k=0}^K (N/n_k) w_k^{*2}\left\{4r_k^2 f_k^2(\theta_k)\right\}^{-1}\left\{1 - r_k^2(2\tau - 1)^2\right\} = \mathcal{O}_p(1)$ and (iii)*

$$\left\{Avar(\widehat{\theta}_{est})\right\}^{-1/2}\left(\widehat{\theta}_{\mathrm{est}} - \theta_0\right) \xrightarrow{d} \mathcal{N}(0, 1), \quad \text{with } Avar(\widehat{\theta}_{est}) := \sum_{k=0}^K \frac{w_k^{*2}}{n_k} \frac{1 - r_k^2(2\tau - 1)^2}{4r_k^2 f_k^2(\theta_k)}.$$

*where $w_k^*$ is the oracle weight defined in equation (3.5).*

*Further, under Assumptions 1, 2 and assume there exists $b_k \ll N^{-1/2}$ for some $1 \le k \le K$, the asymptotic variance of $\widehat{\theta}_{\mathrm{est}}$ is strictly smaller than that of $\widehat{\theta}_0$.i.e., as $N \to \infty$,*

$$Avar(\widehat{\theta}_{est}) < Avar(\widehat{\theta}_0) := \frac{1 - r_0^2(2\tau - 1)^2}{4r_0^2 f_0^2(\theta_k)n_0}.$$

Theorem 4.2 rigorously establishes the asymptotic normality of $\widehat{\theta}_{\mathrm{est}}$ in the absence of unfavorable source sites whose biases are of order $N^{-1/2}$. It is shown that larger $r_k$ or smaller $b_k$ yield larger optimal weights $w_k^*$ and hence a smaller asymptotic variance. Larger $r_k$, however, correspond to weaker privacy guarantees (Proposition 4.1), reflecting the fundamental accuracy–privacy trade-off inherent in the weighted estimator.

The asymptotic variance comparison in Theorem 4.2 further indicates that if at least one source site has a bias smaller than the order $N^{-1/2}$, the weighted estimator achieves a strictly smaller asymptotic variance than $\widehat{\theta}_0$. Hence, even if all the remaining $r_j$'s ($j \neq k$) shrink to zero, the convergence rate of $\widehat{\theta}_{\mathrm{est}}$ is then at least $\left(n_0 r_0^2 + n_k r_k^2\right)^{-1/2}$. As a result, the proposed method improves efficiency and yields narrower confidence intervals than relying solely on the target site data. Intuitively speaking, since the target-site estimator converges at rate $N^{-1/2}$, any source-site bias of smaller order becomes negligible relative to the asymptotic variance; incorporating information from such sites effectively enlarges the usable sample size and turns mild heterogeneity into a blessing.

In addition, it should be noted that our inference is constructed using normal quantiles, which yields narrower confidence intervals compared to self-normalization methods (Liu et al., 2023) at the same confidence level, even without data integration.

Finally, we summarize the asymptotic behaviour of the proposed conservative estimator $\widehat{\theta}_{\mathrm{cons}}$.

**Theorem 4.3** (The property of $\widehat{\theta}_{\mathrm{cons}}$)**.** *Under Assumption A.4, further assume that $\lambda$ is bounded away from 0, $\mathcal{C} \to \infty$ and $\mathcal{C}\sqrt{\widehat{\sigma}_k^2 - \sigma_k^2} = \mathcal{O}_p(1)$ for $0 \le k \le K$ with $\sigma_k^2 = \left\{4r_k^2 f_k^2(\theta_k)\right\}^{-1}\left\{1 - r_k^2(2\tau - 1)^2\right\}$, then we have $\sqrt{N/\widehat{\sigma}_{\mathrm{cons}}^2}(\widehat{\theta}_{\mathrm{cons}} - \theta_0) \xrightarrow{d} \mathcal{N}(0, 1)$.*

Compared with Theorem 4.2, Theorem 4.3 imposes no additional bias-scale restrictions, allowing source sites whose biases are of order $N^{-1/2}$. A closer inspection of the estimator $\widehat{\theta}_{\mathrm{cons}}$ shows that, regardless of the magnitude of $b_k$, the adjusted term $\widetilde{b}_k$ dominates $N^{-1/2}$ due to $C \to \infty$. This construction effectively prevents unfavorable source sites with bias of order $N^{-1/2}$ from affecting the transfer step, ensuring that the conservative method remains robust in more heterogeneous settings.

## 5 EXPERIMENTS

### 5.1 SIMULATION SETUP

In this section, we examine the finite-sample performance of the proposed data integration method on both synthetic and real data. In synthetic data, we fix quantile levels at $\tau \in \{0.25, 0.5, 0.75\}$. Data at each site are generated from either the Normal distribution $\mathcal{N}(\mu_k, 1)$ or the Cauchy distribution $\mathcal{C}(\mu_k, 1)$ with the target site fixed at $\mu_0 = 0$. We set the number of sites as $K = 3$ and the response rate to $r_k = 0.5$ for $0 \le k \le K$, yielding (1.10, 0)-LDP guarantees. The target sample size is $n_0 \in \{20000, 200000\}$, and each source sample size is three times larger. The number of local chains $M_k$ varies between 8 and 20. The learning rate for chain $m$ at site $k$ in the $t$-th iteration is set as $\eta_{k,t_m} = 1/t_m^{0.6}$. Each experiment is replicated $R = 1,000$ times. We consider the following estimators for comparison:

- **ADP(1):** The proposed adaptive weighted estimator with $\lambda = 1$.

- **ADP(decay):** The proposed adaptive weighted estimator with $\lambda = 5\log N/\sqrt{N}$, which satisfies Assumption 1 when the shifts $b_k$ are of constant order.

- **ADP(cons):** The proposed conservative weighted estimator with $\lambda = 1$ and $\mathcal{C} = 1.96$.

- **IVW:** The inverse variance weighted estimator, obtained as ADP with $\lambda = 0$, which ignores parameter shift.

- **Target (Liu et al., 2023):** An ASGD-based estimator using only the target site's data under potential LDP constraints. Inference is conducted via self-normalization.

- **DPSGD** (Song et al., 2013): Differentially private SGD, which adds noise directly to the stochastic gradients rather than using randomized response; see Appendix B.1 for details.

Let $\theta^{(r)}$ denote the estimator in the $r$-th replication, and let $\text{CI}^{(r)}$ represent the corresponding 95% confidence interval (CI). We evaluate the estimators using (i) the mean squared error on the log scale (log MSE): $\log(R^{-1}\sum_{r=1}^{R}|\theta^{(r)} - \theta_0|^2)$; (ii) the empirical coverage probability t the 95% nominal level (ECP): $R^{-1}\sum_{r=1}^{R}\mathbf{1}\{\theta_0 \in \text{CI}^{(r)}\}$; and (iii) the averaged confidence interval length (length of CI): $R^{-1}\sum_{r=1}^{R}|\text{CI}^{(r)}|$. Complete implementation details can be found in Appendix B.1.

## 5.2 Simulation results

First, we evaluate finite-sample performance under scenarios of either vanishing or distinguishable bias, in order to verify Theorem 4.2. Specifically, for each $1 \le k \le 3$, we set $\mu_k$ to be either 0, $0.1/\sqrt{n_0}$, or $100/\sqrt{n_0}$. This setup creates 9 distinct bias levels ranging from complete homogeneity (level 1) to strong heterogeneity (level 9); see Table B.1 for detailed descriptions. Results are presented in Figure 2, B.1, B.2, and B.7(ii). We find that, first, with a fixed bias level, ADP methods yield decreasing log-MSE and CI width as $n_0$ grows, while ECP remains at or above 95%. In addition, ADP methods consistently outperform Target, with uniformly smaller log-MSE and narrower CIs. These empirical results align well with the theoretical guarantees in Theorem 4.2. Compared to competitors, IVW fails under large bias due to ignoring parameter shifts, and DPSGD produces larger MSE and wider CIs. These results illustrate the advantages of our approach.

Next, we examine the finite-sample performance under continuously varying bias values to verify Theorem 4.3. Specifically, for each $1 \le k \le 3$, we set $\mu_k = \mu$ and let $\mu$ vary from $\exp(-5)$ to 1. The results are summarized in Figure 2 and B.3. We observe that ADP(1) and ADP(decay) both deteriorate over a certain bias range, with MSE exceeding that of the Target estimator and ECP dropping far below 95%. In contrast, ADP(cons) remains uniformly robust. These findings demonstrate the usefulness of the conservative weighting strategy.

To further strengthen our simulation study, we conducted a set of sensitivity and robustness analyses, including (i) varying the SGD learning-rate hyperparameter $\beta$; (ii) varying the response rates $r_k$; (iii) varying the number of sites $K$; (iv) using smaller numbers of chains $M_0$; and (v) considering more stringent privacy scenarios where $r_k = 0.25$ for all sites. The results are consistently encouraging. Detailed findings are provided in Appendix B.1.

## 5.3 Real data

We evaluate our method on a real-world dataset widely employed in privacy research: the Government Salary Dataset (Plečko et al., 2024). This dataset is derived from the 2018 American Community Survey conducted by the U.S. Census Bureau and contains over 200,000 records with annual income (USD) as the response. Because income is sensitive personal financial information (Gillenwater et al., 2021), we treat it as privacy-protected data. To reflect the dataset's geographic structure, we partitioned the data by "economic region", treating each region as a site. The original data include nine regions: we select one region (sample size 27,387) as the target site and use three larger regions (Southeast, Far West, and Mideast) as source sites. For all sites, we fix the number of chains at $M_k = 10$, all other hyperparameters follow Section 5. We log-transform the response for estimation and back-transform all reported quantities.

We target quantile levels $\tau \in \{0.25, 0.5\}$ and consider two privacy settings: (1) a homogeneous truthful response rate $r$ for all sites, and (2) heterogeneous rates uniformly distributed on $[0.7, 1.0]$. We report point estimates (Est) and confidence-interval lengths (CI Len) for our two main approaches, ADP(decay) and ADP(cons), and include the Target estimator for reference. The results are summarized in Table 1. As expected, the Target estimator produces the longest intervals, while ADP(decay) yields the shortest in most cases. When privacy constraints are relaxed, the intervals for both ADP(decay) and ADP(cons) become shorter, consistent with our simulation findings. Across all scenarios, the confidence intervals of the Target estimator and those of the proposed ADP methods substantially overlap, indicating good practical performance on real data.

## 6 Concluding Remark

In summary, we propose a unified, privacy-aware framework leveraging auxiliary data to enhance quantile estimation and inference under local differential privacy. By optimally weighting estimators

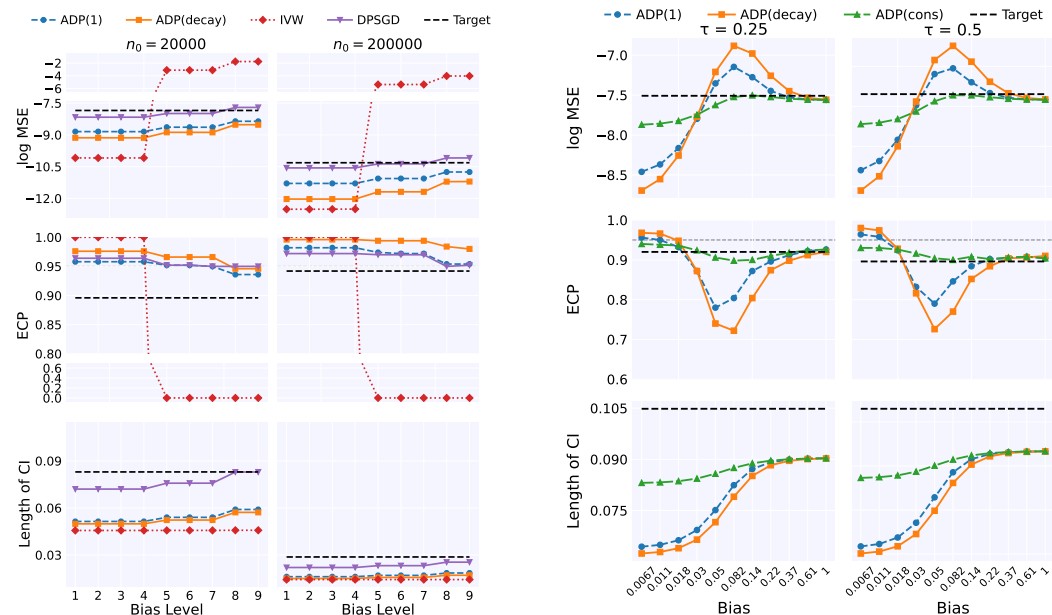

Figure 2: MSEs on the log scale ($\downarrow$), empirical coverage probabilities at the 95% nominal level and the averaged confidence interval lengths ($\downarrow$) under different heterogeneity scenarios. The left panel considers scenarios of either vanishing or distinguishable bias. Bias levels range from complete homogeneity (level 1) to strong heterogeneity (level 9). The quantile level is fixed at $\tau = 0.5$. The right panel considers continuously varying bias values from $\exp(-5)$ to 1. The target sample size is fixed at $n_0 = 20,000$. Data for both panels are generated from normal distributions, with the response rate fixed at $r_k = 0.5$ for all sites.

| Metric | $\tau = 0.25$ | | | $\tau = 0.5$ | | |
|---|---|---|---|---|---|---|
| | ADP(decay) | ADP(cons) | Target | ADP(decay) | ADP(cons) | Target |
| | | | $r_k = 0.7$ | | | |
| Est | 24909 | 23721 | 27442 | 44281 | 43403 | 45374 |
| CI Len | 1992 | 1910 | 3014 | 1041 | 1459 | 1950 |
| | | | hetero $r_k$ | | | |
| Est | 24891 | 23724 | 27442 | 44268 | 44620 | 45374 |
| CI Len | 1753 | 1896 | 3014 | 1009 | 1410 | 1950 |

Table 1: Real-data analysis under different target quantile and truthful response rates. The number of chains per site is fixed at $M_k = 10$. $r_k = 0.7$ denotes a common truthful-response rate across sites, while "hetero $r_k$" denotes site-specific rates range from $0.7$ to $1.0$.

via a PSGD algorithm and penalizing parameter shift, our approach systematically reduces variance and ensures robustness through a conservative alternative. Theoretical results establish consistency and asymptotic normality across diverse heterogeneity settings, demonstrating improved efficiency and reliability over target-only methods, thus offering a principled and practical solution for privacy-preserving quantile inference. However, there also exist some limitations in our framework. First, our asymptotic theory requires the number of chains $M_k$ to diverge; thus, additional investigation into fixed-chain scenarios or non-asymptotic results is necessary. Second, the current framework assumes a fixed number of sites $(K)$; extending the methodology to accommodate diverging $K$ is an interesting open problem. In particular, the strong Gaussian approximation result employed in Theorem 4.1 for a fixed $k$ can be further strengthened to hold uniformly for all $1 \le k \le K_n$, provided that $K_n$ does not diverge too quickly with $n$. Nevertheless, a gap remains in establishing the asymptotic normality of the two proposed estimators in Theorems 4.2–4.3 under diverging $K_n$. In addition, extending the framework to obtain non-asymptotic coverage guarantees is also an important direction for future research.

## REPRODUCIBILITY STATEMENT

All numerical experiments and real-data analyses are fully reproducible via the code included in the submitted anonymized supplementary materials.

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

## A  COMPLETE ASSUMPTIONS AND THEORETICAL RESULTS

**Remark 1.** *Assumption 1 specifies conditions on the tuning parameter $\lambda$. Assumption 2 allows the parameter shift to either vanish faster than $N^{-1/2}$ or be significantly larger than $N^{-1/2}$, covering a wide range of heterogeneous scenarios. This condition relaxes existing assumptions in the literature, which typically require either zero parameter shift (Zhu et al., 2021; Gu & Chen, 2023) or a parameter shift bounded away from zero (Li et al., 2022a; Han et al., 2025).*

**Assumption A.1** (Property of $f_k$). *For every $0 \le k \le K$, $f_k(\cdot)$ is continuous and $f_k(\theta_k) > 0$. In addition, $|f'_k(\cdot)|$ is uniformly bounded by $C$ for some constant $C > 0$.*

**Assumption A.2** (Decaying learning rate). *The learning rate $\eta_{k,t}$ in equation (3.6) satisfies $\eta_{k,t} \asymp t^{-\beta}$ for some constant $\beta \in (1/2, 1)$.*

**Assumption A.3** (Number of chains). *Assume that $M_k^{(\beta+1/2)\wedge(2-\beta)} \lesssim N^{(\beta-1/2)\wedge(1-\beta)}$.*

**Remark 2.** *Assumption A.1 imposes standard regularity conditions on the probability density function. Assumption A.2 requires decay learning rate, commonly assumed in the SGD literature (Polyak & Juditsky, 1992; Lee et al., 2022; Li et al., 2022b). Assumption A.3 restricts the growth rate of the number of PSGD chains $M_k$ relative to the sample size, ensuring accuracy of the final averaged estimator.*

It is worth noting that our theory is not restricted to estimators produced by the PSGD algorithm; rather, it applies broadly to any procedure that provides consistent variance estimation. This requirement is summarized in Assumption A.4.

**Assumption A.4** (Regularity of estimators). *For each $0 \le k \le K$, the estimators $\widehat{\sigma}_k^2$ and $\widehat{\theta}_k$ satisfies that $\widehat{\sigma}_k^2 - \sigma_k^2 = o_p(1)$, and $\sqrt{n_k}(\widehat{\theta}_k - \theta_k) \xrightarrow{d} \mathcal{N}(0, \sigma_k^2)$ with $\sigma_k^2 = \{4r_k^2 f_k^2(\theta_k)\}^{-1}\{1 - r_k^2(2\tau - 1)^2\}$.*

**Remark 3.** *Assumption A.4 is a general regularity condition that accommodates various estimators, including for example, the proposed PSGD method under Assumptions A.1 – A.3, and the classical order statistics-based estimator in the non-private case (Van der Vaart, 2000).*

## B  ADDITIONAL DISCUSSION AND RESULTS

---
**Algorithm B.1** Locally Randomized Compare (Liu et al., 2023)

---
**Require:** Inquiry $\theta$, response rate $r$, private data $x$
**Ensure:** A randomized binary response
 1: Sample $u \sim$ Bernoulli$(r)$
 2: Sample $v \sim$ Bernoulli$(0.5)$
 3: **if** $u = 1$ **then**
 4:     **return** $1_{\theta > x}$
 5: **else**
 6:     **return** $v$
 7: **end if**

---

### B.1  OTHER RESULTS IN SECTION 5 AND COMPLETE EXPERIMENTAL DETAILS.

**Experimental details.**  We provide here additional experimental details not described in the main text. For all experiments, we use 36 Intel(R) Xeon(R) Gold 6271 CPUs, equipped with a total of 128GB of RAM and 500GB of storage. The experiments are implemented using Python 3.12, and the computational time required to generate each figure is approximately 3 to 8 hours.

---

**Algorithm B.2** Conservative Weighting Estimator

---

**Input:** Learning rates $\{\eta_{k,t}\}$, sample sizes $\{n_k\}$, number of chains $\{M_k\}$, target quantile $\tau$, truthful response rates $\{r_k\}$, tuning parameter $\lambda$, and significance level $\alpha$.

**Output:** $\widehat{\theta}_{\text{est}}$ and $[\widehat{\theta}_{\text{est}} - \mathcal{Z}_{\alpha/2}\widehat{\sigma}_{\text{est}}/\sqrt{N}, \widehat{\theta}_{\text{est}} + \mathcal{Z}_{\alpha/2}\widehat{\sigma}_{\text{est}}/\sqrt{N}]$.

**Initialization:** set $\widehat{\theta}_{k,0}^{(m)} \leftarrow 0$ for all $k, m$.

    **for** $0 \leq k \leq K$ **do**

        **for** $1 \leq m \leq M_k$ **do**

            **for** $1 \leq t \leq \lfloor n_k/M_k \rfloor$ **do**

                Obtain the locally randomizer $\widehat{\zeta}_{k,t}^{(m)} = \text{LRC}\left(\widehat{\theta}_{k,t}^{(m)}, r_k, X_{k,t+1}^{(m)}\right)$ using Algorithm B.1.

                Compute $\widehat{\theta}_{k,t}^{(m)}$ according to equation (3.6).

            **end for**

    Compute the chain-specific estimator by $\widehat{\theta}_k^{(m)} = (\lfloor n_k/M_k \rfloor)^{-1} \sum_{t=1}^{\lfloor n_k/M_k \rfloor} \widehat{\theta}_{k,t}^{(m)}$.

        **end for**

    Compute the final estimator and corresponding variance estimator at site $k$ by

$\widehat{\theta}_k = M_k^{-1} \sum_{m=1}^{M_k} \widehat{\theta}_k^{(m)}, \quad \widehat{\sigma}_k^2 = \frac{1}{M_k-1} \sum_{m=1}^{M_k} \lfloor n_k/M_k \rfloor (\widehat{\theta}_k^{(m)} - \widehat{\theta}_k)^2$.

**end for**

Compute the conservative weighted estimator and its variance estimator by $\widehat{\theta}_{\text{cons}} = \sum_{k=0}^{K} \widetilde{w}_k \widehat{\theta}_k$

and $\widehat{\sigma}_{\text{cons}}^2 = \sum_{k=0}^{K} N \widetilde{w}_k \widehat{\sigma}_k^2 / n_k$, where $\widetilde{w}_k = \left\{ \sum_{j=0}^{K} \left(\frac{\widehat{\sigma}_j^2}{n_j} + \lambda \widetilde{b}_j^2\right)^{-1} \right\}^{-1} \left(\frac{\widehat{\sigma}_k^2}{n_k} + \lambda \widetilde{b}_k^2\right)^{-1}$ and $N = \sum_k n_k$.

---

**DPSGD (Song et al., 2013).** Differentially private SGD, which adds noise directly to the stochastic gradients rather than using randomized response. Specifically, we modify the update in (3.3) to

$$\widehat{\theta}_{k,t+1} = \widehat{\theta}_{k,t} - \eta_{k,t} \left\{ \tau - \mathbf{1}\left(X_{k,t+1} \leq \widehat{\theta}_{k,t}\right) + Z_{t+1} \right\},$$

where $Z_t^k \sim \text{Laplace}(0, b)$ with scale $b = 1/\log\{(1 + r_k)/(1 - r_k)\}$. Here we set $\lambda = 1$.

| Level | Description of source sites | Biases $b_k$, $k = 1, \ldots, 3$ |
|---|---|---|
| 1 | All source sites unbiased | $b_1 = b_2 = b_3 = 0$ |
| 2 | One weakly biased source | $b_1 = b_2 = 0,\ b_3 = 0.1/\sqrt{n_0}$ |
| 3 | Two weakly biased sources | $b_1 = 0,\ b_2 = b_3 = 0.1/\sqrt{n_0}$ |
| 4 | Three weakly biased sources | $b_1 = b_2 = b_3 = 0.1/\sqrt{n_0}$ |
| 5 | One strongly biased source | $b_1 = b_2 = 0,\ b_3 = 100/\sqrt{n_0}$ |
| 6 | One weak + one strong | $b_1 = 0,\ b_2 = 0.1/\sqrt{n_0},\ b_3 = 100/\sqrt{n_0}$ |
| 7 | Two weak + one strong | $b_1 = b_2 = 0.1/\sqrt{n_0},\ b_3 = 100/\sqrt{n_0}$ |
| 8 | Two strongly biased sources | $b_1 = 0,\ b_2 = b_3 = 100/\sqrt{n_0}$ |
| 9 | One weak + two strong | $b_1 = 0.1/\sqrt{n_0},\ b_2 = b_3 = 100/\sqrt{n_0}$ |

Table B.1: Bias levels for source sites. Target site has $b_0 = 0$.

## B.2 SENSITIVITY AND ROBUSTNESS ANALYSIS

Unless otherwise specified in this subsection, the target sample size is fixed at $n_0 = 20{,}000$, and the data are generated from normal distributions with a fixed response rate $r_k = 0.5$ for all sites. All other settings follow Section 5.1.

**Sensitivity analysis.** First, we investigate the effect of the learning-rate schedule. Table B.2 summarizes the ECPs. We find that the ECPs remain comparable across different values of $\beta$, indicating that the proposed estimator is stable with respect to the choice of the learning-rate schedule.

Next, we conduct a sensitivity analysis on the truthful response rate. We focus here on a homogeneous setting by fixing $\mu_k = 0$ for all $0 \leq k \leq 3$. We consider three target response rates, $r_0 \in 1, , 0.4, , 0.25$, corresponding respectively to no privacy, moderate local privacy, and strong

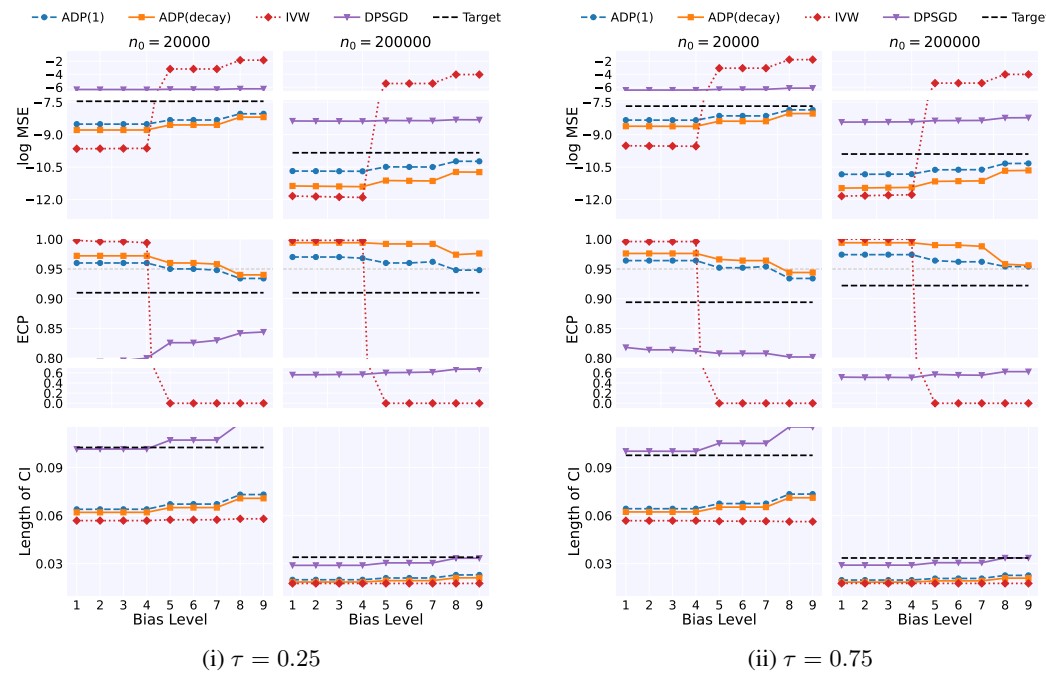

Figure B.1: MSEs on the log scale ($\downarrow$), empirical coverage probabilities at the 95% nominal level and the averaged confidence interval lengths ($\downarrow$) under scenarios of either vanishing or distinguishable bias. Bias levels range from complete homogeneity (level 1) to strong heterogeneity (level 9). Data for both panels are generated from normal distributions, with the response rate fixed at $r_k = 0.5$ for all sites.

local privacy at the target site. For each target scenario, we vary the source-site response rates from 0.25 to 0.9, while keeping all other settings identical to Section 5.1. The results are summarized in Figure B.4.

We observe the following patterns. First, in almost all settings, the ADP estimators consistently improve upon the Target baseline, yielding smaller MAE, narrower confidence intervals, and empirical coverage close to (or above) the nominal 95% level. The improvement becomes more pronounced as the source response rate increases. In addition, when the target site is non-private ($r_0 = 1$), the benefit of integrating source sites is quite limited when their response rates are relatively small.

**Robustness analysis.** First, we vary the number of sites $K$, thereby examining the estimator's performance when incorporating information from fewer or more auxiliary sources. Second, we evaluate the impact of using smaller numbers of local chains $M_0$ to assess sensitivity with respect to variance estimation. Third, we investigate more stringent local-privacy scenarios by setting $r_k = 0.25$ for all sites, corresponding to strong privacy protection across the entire system. Detailed results are provided in Figures B.5, B.6, and B.7(i). Overall, the findings are qualitatively consistent with those in the main text, further demonstrating the robustness of the proposed method.

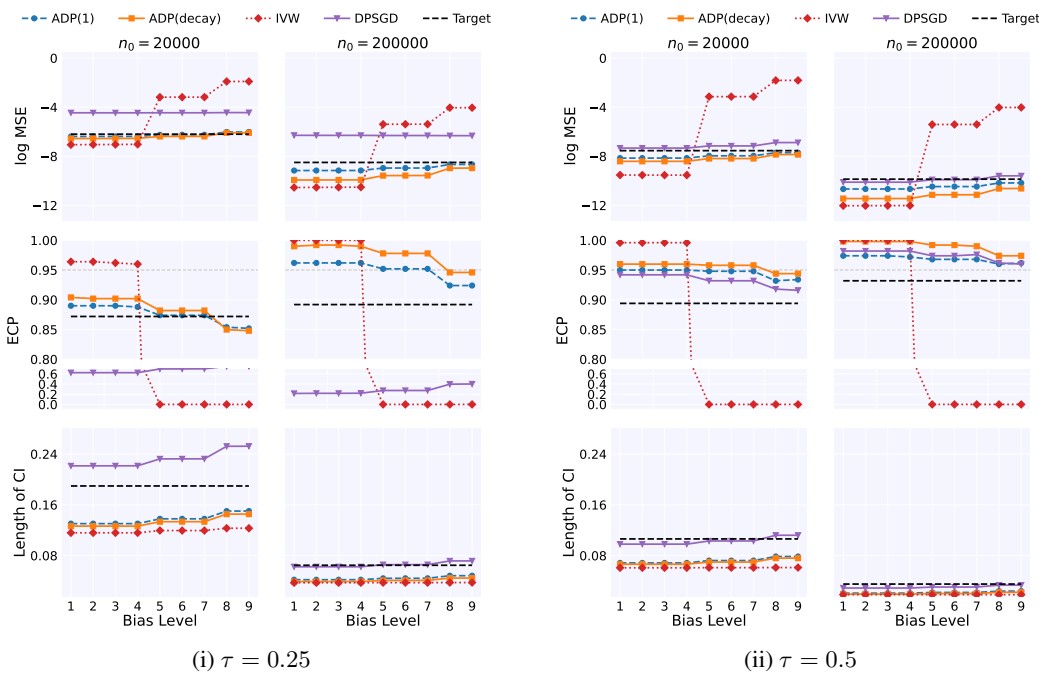

Figure B.2: MSEs on the log scale ($\downarrow$), empirical coverage probabilities at the 95% nominal level and the averaged confidence interval lengths ($\downarrow$) under scenarios of either vanishing or distinguishable bias. Bias levels range from complete homogeneity (level 1) to strong heterogeneity (level 9). Data for both panels are generated from Cauchy distributions, with the response rate fixed at $r_k = 0.5$ for all sites.

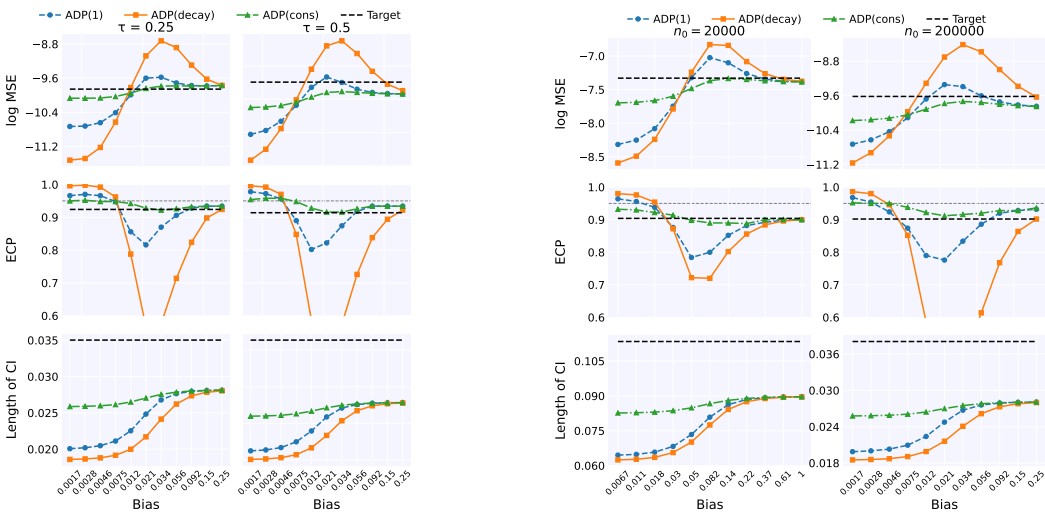

Figure B.3: MSEs on the log scale ($\downarrow$), empirical coverage probabilities at the 95% nominal level, and averaged confidence interval lengths ($\downarrow$) under continuously varying bias values. The left panel fixes the target sample size at $n_0 = 20,000$, and the right panel fixes the quantile level at $\tau = 0.75$. Data for both panels are generated from normal distributions, with the response rate fixed at $r_k = 0.5$ for all sites.

| $\beta$ | Method | 0.007 | 0.011 | 0.018 | 0.030 | 0.135 | 0.223 | 0.368 | 0.607 | 1.000 |
|---------|--------|-------|-------|-------|-------|-------|-------|-------|-------|-------|
| | ADP(1) | 96.9 | 96.2 | 93.6 | 85.8 | 88.2 | 91.0 | 92.0 | 92.1 | 92.3 |
| 0.65 | ADP(cons) | 94.0 | 94.2 | 93.8 | 93.2 | 91.2 | 91.7 | 92.0 | 92.2 | 92.4 |
| | ADP(decay) | 98.4 | 97.6 | 94.2 | 83.4 | 84.2 | 88.6 | 90.4 | 91.0 | 91.2 |
| | ADP(1) | 96.1 | 95.6 | 94.0 | 87.5 | 87.4 | 90.6 | 91.8 | 91.8 | 91.7 |
| 0.70 | ADP(cons) | 94.0 | 93.9 | 93.6 | 92.9 | 90.9 | 91.8 | 91.7 | 91.7 | 91.7 |
| | ADP(decay) | 98.0 | 97.8 | 94.4 | 86.6 | 81.8 | 87.6 | 89.8 | 91.0 | 91.6 |
| | ADP(1) | 95.6 | 95.6 | 94.5 | 89.9 | 85.3 | 89.1 | 90.5 | 91.3 | 91.5 |
| 0.75 | ADP(cons) | 93.8 | 93.6 | 93.6 | 93.0 | 90.0 | 90.2 | 90.6 | 91.3 | 91.4 |
| | ADP(decay) | 98.0 | 97.8 | 96.2 | 90.6 | 79.2 | 84.8 | 88.4 | 89.0 | 89.8 |

Table B.2: Empirical 95% coverage probabilities under bias values ranging from $\exp(-5)$ to 1, across varying learning rate parameters $\beta$. The target sample size is fixed at $n_0 = 20{,}000$, with the target quantile set to $\tau = 0.5$. Data are generated from normal distributions with a fixed response rate of $r_k = 0.5$ across all sites.

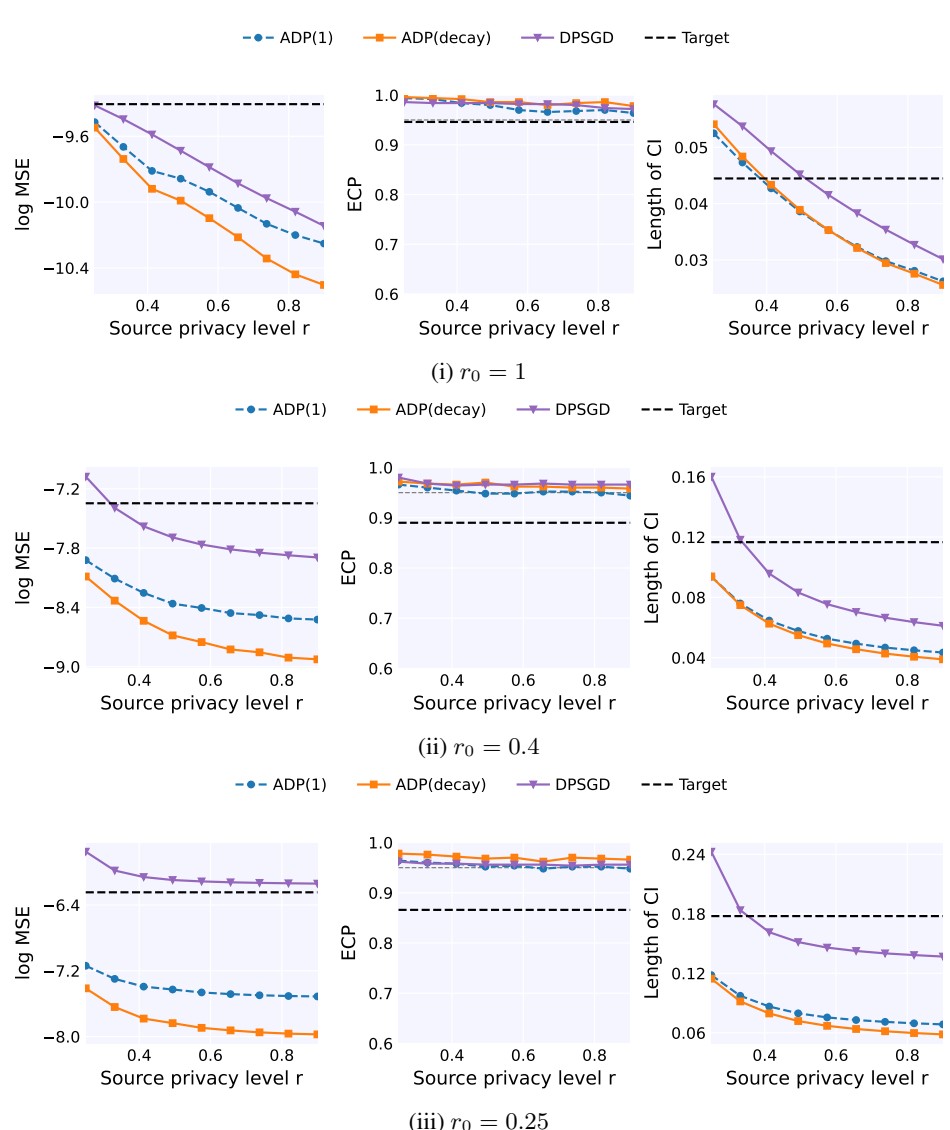

Figure B.4: MSEs on the log scale (↓), empirical coverage probabilities at the 95% nominal level, and averaged confidence interval lengths (↓) under varying truthful response rates $r_k = r$ for the source site. The target sample size is fixed at $n_0 = 20{,}000$, with the target quantile set to $\tau = 0.5$. Data are generated from normal distributions with a fixed $\mu_k = 0$ across all sites.

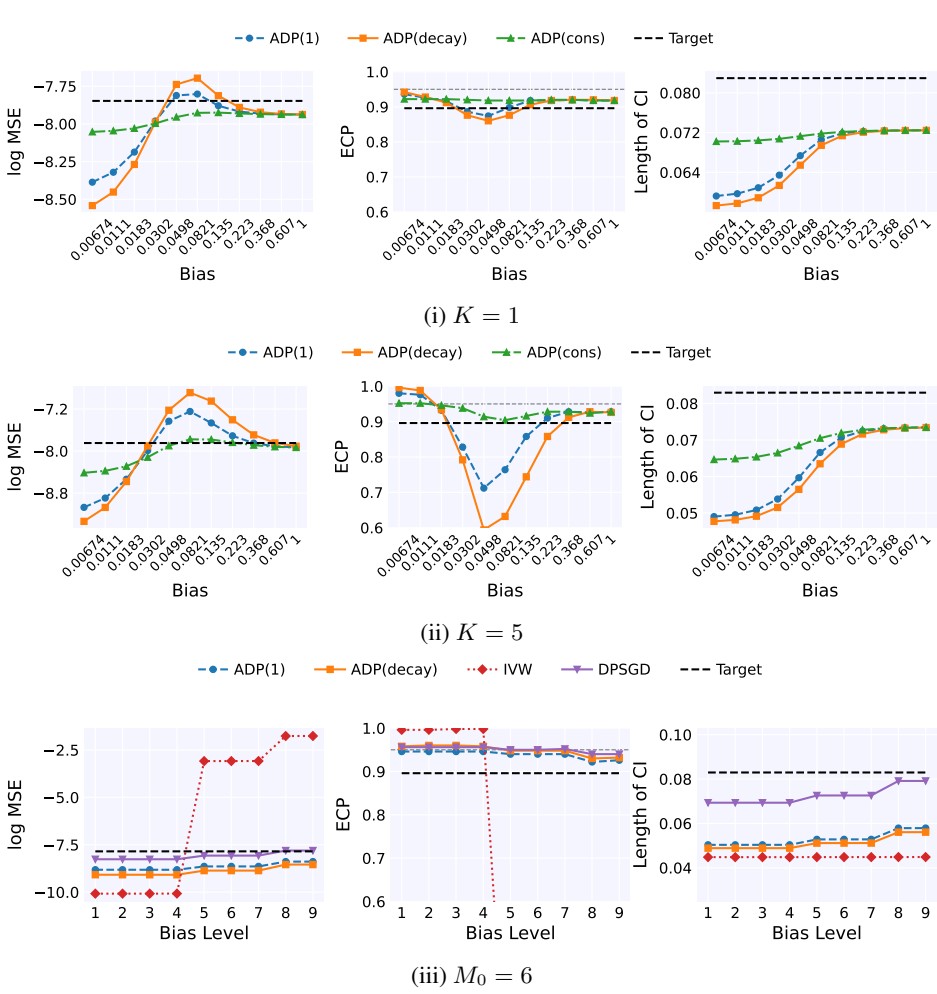

Figure B.5: MSEs on the log scale ($\downarrow$), empirical coverage probabilities at the 95% nominal level, and averaged confidence-interval lengths ($\downarrow$) under different heterogeneity scenarios. The target sample size is fixed at $n_0 = 20{,}000$, with the target quantile set to $\tau = 0.5$. Data are generated from Normal distributions with a fixed response rate of $r_k = 0.5$ across all sites.

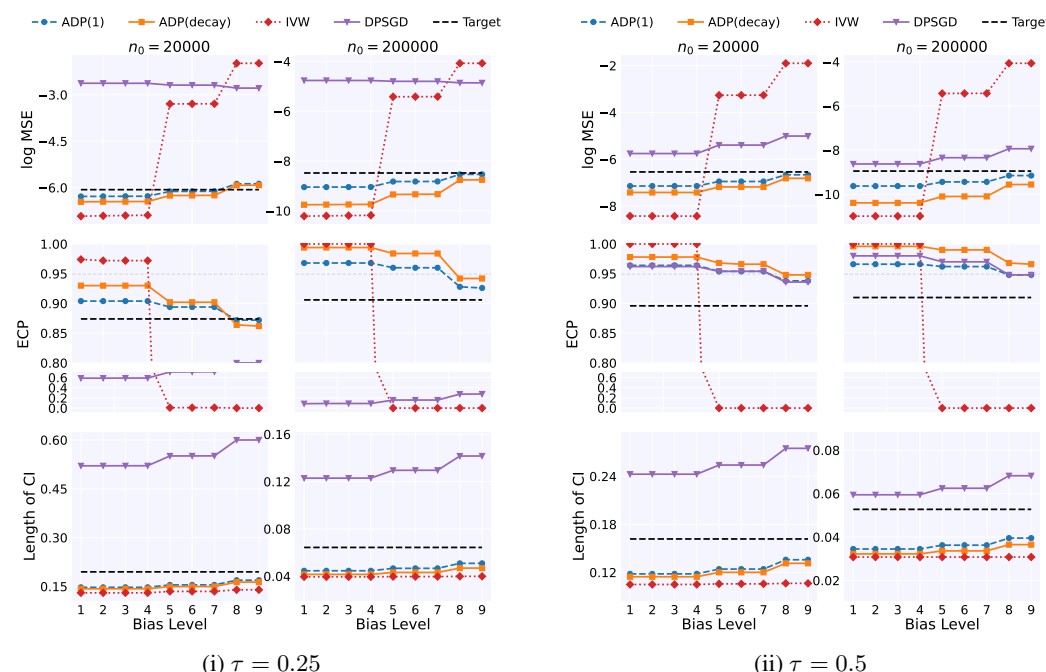

(i) $\tau = 0.25$          (ii) $\tau = 0.5$

Figure B.6: MSEs on the log scale ($\downarrow$), empirical coverage probabilities at the 95% nominal level and the averaged confidence interval lengths ($\downarrow$) under scenarios of either vanishing or distinguishable bias. Bias levels range from complete homogeneity (level 1) to strong heterogeneity (level 9). Data for both panels are generated from normal distributions, with the response rate fixed at $r_k = 0.25$ for all sites.

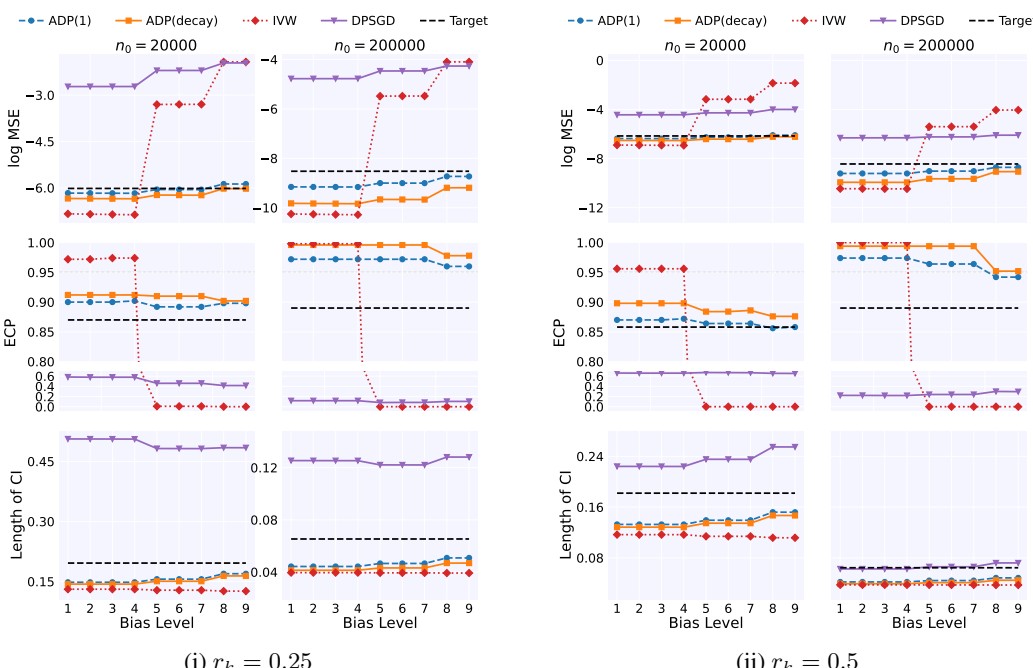

(i) $r_k = 0.25$          (ii) $r_k = 0.5$

Figure B.7: MSEs on the log scale ($\downarrow$), empirical coverage probabilities at the 95% nominal level and the averaged confidence interval lengths ($\downarrow$) under scenarios of either vanishing or distinguishable bias. Bias levels range from complete homogeneity (level 1) to strong heterogeneity (level 9). Data for the left panels are generated from the normal distribution, and those for the right panels are generated from the Cauchy distribution.

## C  TECHNICAL LEMMAS

**Notations.**  We first introduce some notation used throughout the paper. For two positive sequences $a_n$ and $b_n$, write $a_n \ll b_n$ or $a_n = \mathrm{o}(b_n)$ if $a_n/b_n \to 0$ as $n \to \infty$. Similarly, write $a_n \lesssim b_n$ or $a_n = O(b_n)$ if there exists a constant $C < \infty$ such that $a_n/b_n \leq C$ for all sufficiently large $n$. Moreover, write $a_n \asymp b_n$ if both $a_n \lesssim b_n$ and $b_n \lesssim a_n$ hold simultaneously. For two random sequences $\{X_n\}$ and $\{Y_n\}$, write $X_n = \mathrm{o}_p(Y_n)$ if $P(|X_n/Y_n| > \varepsilon) \to 0$ for any $\varepsilon > 0$. Write $X_n = O_p(Y_n)$ if for every $\varepsilon > 0$, there exists a constant $M > 0$ such that $\limsup_{n \to \infty} P(|X_n|/|Y_n| > M) < \varepsilon$. Write $X_n \xrightarrow{d} X$ if the random sequence $X_n$ converges in distribution to a random variable $X$. Finally, $\lfloor x \rfloor$ denotes the largest integer less than or equal to $x$.

**Lemma C.1.** *Denote* $Z_k^{(m)} = \sqrt{(n_k/M_k)}(\widehat{\theta}_k^{(m)} - \theta_k)$. *We have*

$$\mathbb{E}(Z_k^{(m)})^2 - \sigma_k^2 = \mathcal{O}\Big(1/(n_k/M_k)^{(\beta-1/2)\wedge(1-\beta)}\Big).$$

$$\mathbb{E}(Z_k^{(m)}) = \mathcal{O}\Big(1/(n_k/M_k)^{(\beta/2-1/4)\wedge(1/2-\beta/2)}\Big).$$

*Proof.* The proof is shown in Corollary 6 in Gadat & Panloup (2023). $\qquad\square$

**Lemma C.2.** *For each* $0 \leq k \leq K$, *define the weighted estimator as* $\widehat{\theta}(\mathbf{w}) = \sum_{k=0}^{K} w_k \widehat{\theta}_k$, *let* $\theta^*(\mathbf{w}) = \sum_{k=0}^{K} w_k \theta_k$ *and* $\sigma^2(\mathbf{w}) = \sum_{k=0}^{K} \frac{N}{n_k} w_k^2 \sigma_k^2$ *represent its corresponding true parameter and variance, respectively. Then under assumption A.4, we have:*

$$\sqrt{N}\left(\widehat{\theta}(\mathbf{w}) - \theta^*(\mathbf{w})\right) \xrightarrow{d} \mathcal{N}\left(0, \sigma^2(\mathbf{w})\right).$$

*Proof.* Lemma C.2 follows naturally as a corollary of Theorem 4.1. $\qquad\square$

**Lemma C.3.** *For every* $0 \leq k \leq K$, *we have*

$$\mathbb{E}|\widehat{b}_k - b_k| \quad \lesssim \quad \frac{1}{\sqrt{N}} \tag{C.1}$$

$$\mathbb{E}|\widehat{b}_k^2 - b_k^2| \quad \lesssim \quad \frac{1}{N} + |b_k|\frac{1}{\sqrt{N}} \tag{C.2}$$

*Proof.* Denote $Z_k^{(m)} = \sqrt{(n_k/M_k)}(\widehat{\theta}_k^{(m)} - \theta_k)$, $\bar{Z}_k = M_k^{-1} \sum_{m=1}^{M_k} Z_k^{(m)}$ for every $0 \leq k \leq K$.

**(i): Proof of equation** (C.1). Recall that $b_k = \theta_k - \theta_0$ and $\widehat{b}_k = \widehat{\theta}_k - \widehat{\theta}_0$, with $\widehat{\theta}_k = \frac{1}{M_K} \sum_{m=1}^{M_K} \widehat{\theta}_k^{(m)}$. First, note that

$$\mathbb{E}|\widehat{b}_k - b_k| \lesssim \sqrt{\mathbb{E}|\widehat{b}_k - b_k|^2}.$$

It then suffices to analysize $\mathbb{E}|\widehat{b}_k - b_k|^2$. It could be proved that under Assumption A.3

$$\begin{aligned}
\mathbb{E}|\widehat{b}_k - b_k|^2 \quad &\lesssim \quad \mathbb{E}\big(\widehat{\theta}_0 - \theta_0\big)^2 + \mathbb{E}\big(\widehat{\theta}_k - \theta_k\big)^2 = \frac{M_0}{n_0}\{\mathbb{E}^2(\bar{Z}_0) + \mathrm{var}(\bar{Z}_0)\} \\
&+ \frac{M_k}{n_k}\{\mathbb{E}^2(\bar{Z}_k) + \mathrm{var}(\bar{Z}_k)\} \lesssim \frac{M_0}{n_0}\Big(1/(n_0/M_0)^{(\beta-1/2)\wedge(1-\beta)}\Big) \\
&+ \frac{M_k}{n_k}\Big(1/(n_k/M_k)^{(\beta-1/2)\wedge(1-\beta)}\Big) + \frac{1}{N} \lesssim \frac{1}{N}.
\end{aligned}$$

This yields $\mathbb{E}|\widehat{b}_k - b_k| \lesssim 1/\sqrt{N}$.

**(ii): Proof of equation** (C.2). It could be verified that

$$\begin{aligned}
\mathbb{E}|\widehat{b}_k^2 - b_k^2| \quad &\leq \quad \mathbb{E}|\widehat{b}_k - b_k|^2 + 2b_k\mathbb{E}|\widehat{b}_k - b_k| \lesssim \mathbb{E}|\widehat{b}_k - b_k|^2 + |b_k|\sqrt{\mathbb{E}|\widehat{b}_k - b_k|^2} \\
&\lesssim \quad \frac{1}{N} + \frac{|b_k|}{\sqrt{N}}.
\end{aligned}$$

This finishes the whole Lemma proof. $\qquad\square$

**Lemma C.4** (Consistency). *Under Assumption A.4, consider the following three scenarios: (a) $\lambda = 0$, and the bias scale satisfies Assumption 2 (1); (b) $\lambda$ is bounded away from 0, and the bias scale satisfies Assumption 2 (2); (c) $\lambda$ satisfies Assumption 1, and the bias scale satisfies Assumption 2. Then, for each $0 \leq k \leq K$, with probability $1 - o(1)$, we have*

$$|\widehat{w}_k - w_k^*| \lesssim \left\{ \sum_{k=0}^{K} \prod_{j \neq k} \left( \frac{\sigma_j^2}{N} + \lambda b_j^2 \right) \right\}^{-1} \left\{ \sum_{k=0}^{K} \sum_{i \neq k} \left( \frac{\lambda |b_i|}{\sqrt{N}} + \frac{o(1)}{N} \right) \prod_{j \neq i}^{j \neq k} \left( \frac{\sigma_j^2}{N} + \lambda b_j^2 \right) \right\} = o(1).$$

## D  PROOF OF THE MAIN THEORETICAL RESULTS

Without loss of generality, we simply assume that $\lfloor n_k/M_k \rfloor = n_k/M_k =: T_k$.

### D.1  PROOF OF THEOREM 4.1

First, we have $\mathbb{E}|\widehat{b}_k^2 - b_k^2| \lesssim 1/N + |b_k|/\sqrt{N}$ in Lemma C.3. Next, for any fixed $k = 0, \ldots, K$, one rewrites

$$\widehat{\theta}_k - \theta_k = \frac{1}{M_k} \sum_{m=1}^{M_k} \frac{1}{T_k} \sum_{t=1}^{T_k} (\widehat{\theta}_{k,t}^{(m)} - \theta_k) =: \frac{1}{M_k} \sum_{m=1}^{M_k} \mathcal{T}_k^{(m)},$$

where

$$\mathcal{T}_k^{(m)} = \widehat{\theta}_k^{(m)} - \theta_k = \frac{1}{T_k} \sum_{t=1}^{T_k} (\widehat{\theta}_{k,t}^{(m)} - \theta_k).$$

Let $U_{k,t}^{(m)}$ and $V_{k,t}^{(m)}$ be i.i.d. Bernoulli variables, mutually independent and also independent of $X_{k,t}^{(m)}$, with

$$\mathbb{P}(U_{k,t}^{(m)} = 1) = r_k, \quad \mathbb{P}(U_{k,t}^{(m)} = 0) = 1 - r_k, \qquad \mathbb{P}(V_{k,t}^{(m)} = 1) = \mathbb{P}(V_{k,t}^{(m)} = 0) = \tfrac{1}{2}.$$

Then it could be found that equation (3.6) could be rewritten as

$$\widehat{\theta}_{k,t+1}^{(m)} = \widehat{\theta}_{k,t}^{(m)} - \eta_{k,t} G_k(\widehat{\theta}_{k,t}^{(m)}, \zeta_{k,t+1}^{(m)}),$$

where $\zeta_{k,t}^{(m)} = \left( X_{k,t}^{(m)}, U_{k,t}^{(m)}, V_{k,t}^{(m)} \right)^{\top}$, and

$$G_k(\theta, \zeta_{k,t}^{(m)}) = \frac{1 + r_k - 2r_k\tau}{2} \left\{ \mathbf{1}(X_{k,t}^{(m)} \leq \theta) U_{k,t}^{(m)} + (1 - U_{k,t}^{(m)})(1 - V_{k,t}^{(m)}) \right\}$$

$$- \frac{1 - r_k + 2r_k\tau}{2} \left\{ \mathbf{1}(X_{k,t}^{(m)} > \theta) U_{k,t}^{(m)} + (1 - U_{k,t}^{(m)}) V_{k,t}^{(m)} \right\}.$$

Define

$$\varepsilon_{k,t}^{(m)} = g_k(\widehat{\theta}_{k,t-1}^{(m)}) - G_k(\widehat{\theta}_{k,t-1}^{(m)}, \zeta_{k,t}^{(m)}), \quad \widetilde{\varepsilon}_{k,t}^{(m)} = g_k(\theta_k) - G_k(\theta_k, \zeta_{k,t}^{(m)}).$$

Elementary calculation shows that

$$\mathbb{E}(\varepsilon_{k,t}^{(m)2}|\mathcal{F}_{k,t-1}^{(m)}) = \frac{1 + r_k - 2r_k F_k(\widehat{\theta}_{k,t-1}^{(m)})}{2} - \left( \frac{1 + r_k - 2r_k F_k(\widehat{\theta}_{k,t-1}^{(m)})}{2} \right)^2$$

$$= \frac{1 - r_k^2 \left\{ 2F_k(\widehat{\theta}_{k,t-1}^{(m)}) - 1 \right\}^2}{4}$$

$$\xrightarrow{\mathbb{P}} \frac{1 - r_k^2(2\tau - 1)^2}{4},$$

where the convergence in probability holds by the consistency of the quantile estimation and the continuous mapping theorem. Denote $\Delta_{k,t}^{(m)} = \widehat{\theta}_{k,t}^{(m)} - \theta_k$, $H_k = r_k f_k(\theta_k)$, $B_{k,t} = 1 - \eta_{k,t} H_k$, $A_{k,j}^t = \sum_{s=j}^{t} \left( \prod_{i=j+1}^{s} B_{k,i} \right) \eta_{k,i}$ for any $j \leq t$. We decompose that

$$\mathcal{T}_k^{(m)} = \frac{1}{T_k} \sum_{j=1}^{T_k} (\widehat{\theta}_{k,j}^{(m)} - \theta_k) = \frac{1}{T_k} \sum_{j=1}^{T_k} \Delta_{k,j}^{(m)}$$

$$= \frac{1}{T_k} A_{k,0}^{T_k-1} B_{k,0} \Delta_{k,0} + \frac{1}{T_k} \sum_{j=0}^{T_k-1} A_{k,j}^{T_k-1} r_{k,j}^{(m)} + \frac{1}{T_k} \sum_{j=0}^{T_k-1} \left( A_{k,j}^{T_k-1} - H_k^{-1} \right) \varepsilon_{k,j+1}^{(m)}$$

$$+ \frac{1}{T_k} \sum_{j=0}^{T_k-1} H_k^{-1} \left( \varepsilon_{k,j+1}^{(m)} - \widetilde{\varepsilon}_{k,j+1}^{(m)} \right) + \frac{1}{T_k} \sum_{j=0}^{T_k-1} H_k^{-1} \widetilde{\varepsilon}_{k,j+1}^{(m)}$$

$$=: \mathcal{T}_{k,1}^{(m)} + \mathcal{T}_{k,2}^{(m)} + \mathcal{T}_{k,3}^{(m)} + \mathcal{T}_{k,4}^{(m)} + \mathcal{T}_{k,5}^{(m)},$$

in which

$$r_{k,j}^{(m)} = H_k(\widehat{\theta}_{k,j}^{(m)} - \theta_k) - g_k(\widehat{\theta}_{k,j}^{(m)}).$$

**For $\mathcal{T}_{k,1}^{(m)}$:** According to Lemma C.4 of Xie et al. (2024), one has $\left| A_{k,0}^{t-1} \right| \le C_0$ uniformly for all $t \ge 1$. Further observe that $\Delta_{k,0}^{(m)} \equiv \widehat{\theta}_{k,0} - \theta_k$ for all $1 \le m \le M_k$, thus one obtains

$$\left| \frac{1}{M_k} \sum_{m=1}^{M_k} \mathcal{T}_{k,1}^{(m)} \right| = \mathcal{O}\left( M_k/n_k \right).$$

**For $\mathcal{T}_{k,2}^{(m)}$:** Theorem 5 of Gadat & Panloup (2023) shows that

$$\max_{1 \le m \le M_k} \mathbb{E} \left| \widehat{\theta}_{k,t}^{(m)} - \theta_k \right|^2 \lesssim \eta_{k,t}.$$

According to the uniform boundedness of $\left| A_{k,j}^{t-1} \right|$ and the fact that $|r_{k,t}^{(m)}| \lesssim |\widehat{\theta}_{k,t}^{(m)} - \theta_k|^2$,

$$\mathbb{E} \left| \frac{1}{M_k} \sum_{m=1}^{M_k} \mathcal{T}_{k,2}^{(m)} \right| \lesssim \mathbb{E} \frac{1}{M_k} \sum_{m=1}^{M_k} \frac{1}{T_k} \sum_{t=0}^{T_k-1} \left| r_{k,t}^{(m)} \right|$$

$$\le \frac{1}{M_k} \sum_{m=1}^{M_k} \frac{1}{T_k} \sum_{t=0}^{T_k-1} \mathbb{E} |\widehat{\theta}_{k,t}^{(m)} - \theta_k|^2$$

$$= \mathcal{O}\left( (M_k/n_k)^a \right).$$

Hence,

$$\left| \frac{1}{M_k} \sum_{m=1}^{M_k} \mathcal{T}_{k,2}^{(m)} \right| = \mathcal{O}_p\left( (M_k/n_k)^a \right).$$

**For $\mathcal{T}_{k,3}^{(m)}$:** For any fixed $p > 0$, note that $\max_{1 \le m \le M_k} \mathbb{E}|\varepsilon_{k,j}^{(m)}|^{2p} = \mathbb{E}|\varepsilon_{1,j}^{(m)}|^{2p}$ is bounded. Following the arguments in Xie et al. (2024), one has $\left\| \mathcal{T}_{3,k}^{(m)} \right\|_{2p} = \mathcal{O}\left( (n_k/M_k)^{-1+a/2} \right)$. Then, using the Lemma A in Chapter 9.2.6 of Serfling (2009) and the independence over $m$, one has that

$$\left\| \frac{1}{M_k} \sum_{m=1}^{M_k} \mathcal{T}_{k,3}^{(m)} \right\|_{2p} = \mathcal{O}\left( M_k^{-1/2} (n_k/M_k)^{-1+a/2} \right),$$

which implies that

$$\left| \frac{1}{M_k} \sum_{m=1}^{M_k} \mathcal{T}_{k,3}^{(m)} \right| = \mathcal{O}_p\left( M_k^{-1/2} (n_k/M_k)^{-1+a/2} \right).$$

**For $\mathcal{T}_{k,4}^{(m)}$:** Observe that $\varepsilon_{k,j}^{(m)} - \widetilde{\varepsilon}_{k,j}^{(m)} = g_k(\widehat{\theta}_{k,j-1}^{(m)}) - G_k(\widehat{\theta}_{k,j-1}^{(m)}, \zeta_{k,j}^{(m)}) + G_k(\theta_k, \zeta_{k,j}^{(m)})$, and

$$\mathbb{E} \left( \varepsilon_{k,j}^{(m)} - \widetilde{\varepsilon}_{k,j}^{(m)} \right)^2 \lesssim \mathbb{E} g_k^2(\widehat{\theta}_{k,j-1}^{(m)}) + \mathbb{E} \left\{ G_k(\widehat{\theta}_{k,j-1}^{(m)}, \zeta_{k,j}) - G_k(\theta_k, \zeta_{k,j}) \right\}^2$$

$$\lesssim \mathbb{E} \left| \widehat{\theta}_{k,j-1}^{(m)} - \theta_k \right|^2 + \mathbb{E} \left| \widehat{\theta}_{k,j-1}^{(m)} - \theta_k \right|$$

$$\lesssim \mathbb{E} \left| \widehat{\theta}_{k,j-1}^{(m)} - \theta_k \right|^2 + \left\{ \mathbb{E} \left| \widehat{\theta}_{k,j-1}^{(m)} - \theta_k \right|^2 \right\}^{1/2} \lesssim \eta_{k,t}^{1/2},$$

where the last inequality holds by Theorem 5 of Gadat & Panloup (2023), and the constant does not depend on $m$.

Observe that $\sum_{t=1}^{T_k} H_k^{-1} \left( \varepsilon_{k,t}^{(m)} - \widetilde{\varepsilon}_{k,t}^{(m)} \right)$ is a martingale for each $m$ (independent over $1 \leq m \leq M_k$), Burkholder's inequality entails that

$$\left\| \sum_{t=1}^{T_k} H_k^{-1} \left( \varepsilon_{k,t}^{(m)} - \widetilde{\varepsilon}_{k,t}^{(m)} \right) \right\|_2 \lesssim \left\{ \sum_{t=1}^{T_k} \left\| H_k^{-1} \left( \varepsilon_{k,t}^{(m)} - \widetilde{\varepsilon}_{k,t}^{(m)} \right) \right\|_2^2 \right\}^{1/2}$$

$$\lesssim \left( \sum_{t=1}^{T_k} \eta_{k,t}^{1/2} \right)^{1/2} \lesssim (n_k/M_k)^{\{1-a/2\}/2}.$$

Hence,

$$\left\| \mathcal{T}_{k,4}^{(m)} \right\|_2 = \left\| \frac{1}{T_k} \sum_{j=1}^{T_k} H^{-1} \left( \varepsilon_{k,j}^{(m)} - \widetilde{\varepsilon}_{k,j}^{(m)} \right) \right\|_2 = \mathcal{O} \left( (n_k/M_k)^{-1/2-a/4} \right),$$

which further implies that

$$\frac{1}{M_k} \sum_{m=1}^{M_k} \mathcal{T}_{k,4}^{(m)} = \mathcal{O}_p \left( M_k^{-1/2} (n_k/M_k)^{-1/2-a/4} \right).$$

**For $\mathcal{T}_{k,5}^{(m)}$:** Elementary calculation shows that

$$\mathbb{E} \widetilde{\varepsilon}_{k,j}^{(m)2} = \frac{1 - r_k^2 (2\tau - 1)^2}{4} =: S_k.$$

Applying Theorem 2.6.7 of Csörgo & Révész (1981) with $H(x) = x^{2p}$ and $x_n = n_k^{\beta_0}$, there exist i.i.d. standard normal $\widetilde{Z}_{k,i}$'s and some $a_k, C_k > 0$ (depending on the distribution of $H_k^{-1} \widetilde{\varepsilon}_{k,j}^{(m)}$) such that

$$\mathbb{P} \left( \left| \sum_{m=1}^{M_k} \sum_{t=1}^{T_k} \frac{H_k^{-1} \widetilde{\varepsilon}_{k,t}^{(m)}}{\sqrt{H_k^{-1} S_k H_k^{-1}}} - \sum_{i=1}^{n_k} \widetilde{Z}_{k,i} \right| > n_k^{\beta_0} \right) \leq C_k a_k^{-2p} n_k^{1-2p\beta_0}.$$

Thus,

$$\mathbb{P} \left( \left| \frac{1}{n_k} \sum_{m=1}^{M_k} \sum_{t=1}^{T_k} \frac{H_k^{-1} \widetilde{\varepsilon}_{k,t}^{(m)}}{\sqrt{H_k^{-1} S_k H_k^{-1}}} - \frac{1}{n_k} \sum_{i=1}^{n_k} \widetilde{Z}_{k,i} \right| > n_k^{-1+\beta_0} \right) \lesssim n_k^{1-2p\beta_0}.$$

For $p > 2$, one selects $\beta_0 \in (1/p, 1/2)$, the Borel-Cantelli lemma leads to

$$\left| \frac{1}{M_k} \sum_{m=1}^{M_k} \mathcal{T}_{k,5}^{(m)} - \frac{1}{n_k} \sum_{i=1}^{n_k} Z_{k,i} \right| = \mathcal{O}_{a.s.} \left( n_k^{-1+\beta_0} \right),$$

where $Z_{k,i}$'s are i.i.d. normal r.v.'s with mean zero and covariance $H_k^{-1} S_k H_k^{-1}$.

Therefore, we obtain that

$$\left| \widehat{\theta}_k - \theta_k - \frac{1}{n_k} \sum_{i=1}^{n_k} Z_{k,i} \right| = o_p \left( n_k^{-1/2} \right),$$

which completes the proof of the weak convergence result.

As for the consistency of $\widehat{\sigma}_k^2$, we rewrite

$$\widehat{\sigma}_k^2 = \frac{n_k}{M_k - 1} \frac{1}{M_k} \sum_{m=1}^{M_k} \left( \widehat{\theta}_k^{(m)} - \theta_k \right)^2 - \frac{n_k}{M_k - 1} \left\{ \frac{1}{M_k} \sum_{m=1}^{M_k} \left( \widehat{\theta}_k^{(m)} - \theta_k \right) \right\}^2$$

$$= \frac{n_k}{M_k - 1} \frac{1}{M_k} \sum_{m=1}^{M_k} \mathcal{T}_k^{(m)2} - \frac{n_k}{M_k - 1} \left\{ \frac{1}{M_k} \sum_{m=1}^{M_k} \mathcal{T}_k^{(m)} \right\}^2.$$

Recall the definitions of $\mathcal{T}_{k,j}^{(m)}$ for $1 \leq j \leq 5$.

**For $\mathcal{T}_{k,1}^{(m)}$:** According to Lemma C.4 of Xie et al. (2024), one has $\left| A_{k,0}^{t-1} \right| \leq C_0$ uniformly for all $t \geq 1$. Further observe that $\Delta_{k,0}^{(m)} \equiv \widehat{\theta}_{k,0} - \theta_k$ for all $1 \leq m \leq M_k$, thus one obtains $\left\| \mathcal{T}_{k,1}^{(m)} \right\|_2 = \mathcal{O}\left( T_k^{-1} \right)$, where the constant does not depend on $m$.

**For $\mathcal{T}_{k,2}^{(m)}$:** Consider that

$$\left| \frac{1}{T_k} \sum_{j=0}^{T_k-1} A_{k,j}^{T_k-1} r_{k,j}^{(m)} \right| \lesssim \frac{1}{T_k} \sum_{j=0}^{T_k-1} \left| r_{k,j}^{(m)} \right| \lesssim \frac{1}{T_k} \sum_{j=0}^{T_k-1} \left| \widehat{\theta}_{k,j}^{(m)} - \theta_k \right|^2.$$

Then,

$$\left\| \frac{1}{T_k} \sum_{j=0}^{T_k-1} A_{k,j}^{T_k-1} r_{k,j}^{(m)} \right\|_2 \lesssim \frac{1}{T_k} \sum_{j=0}^{T_k-1} \left\| \widehat{\theta}_{k,j}^{(m)} - \theta_k \right\|_4^2.$$

Applying Theorem 5 of Gadat & Panloup (2023), we have

$$\mathbb{E} \left| \widehat{\theta}_{k,j}^{(m)} - \theta_k \right|^4 \lesssim \eta_{k,j}^2.$$

Since $\eta_{k,j} \asymp j^{-a}$ with $a > 1/2$, it follows that $\left\| \mathcal{T}_{k,2}^{(m)} \right\|_2 = \mathcal{O}\left( T_k^{-1/2} \right)$.

**For $\mathcal{T}_{k,3}$:** As shown in the previous arguments, one has $\left\| \mathcal{T}_{3,k}^{(m)} \right\|_{2p} = \mathcal{O}\left( T_k^{-1+a/2} \right) = \mathcal{O}\left( T_k^{-1/2} \right)$, since $a < 1$.

**For $\mathcal{T}_{k,4}$:** Observe that $\sum_{j=1}^{T_k} H_k^{-1} \left( \varepsilon_{k,j}^{(m)} - \widetilde{\varepsilon}_{k,j}^{(m)} \right)$ is a martingale for each $m$ (independent over $1 \leq m \leq M_k$), Burkholder's inequality entails that

$$\left\| \sum_{j=1}^{T_k} H_k^{-1} \left( \varepsilon_{k,j}^{(m)} - \widetilde{\varepsilon}_{k,j}^{(m)} \right) \right\|_2 \lesssim \left\{ \sum_{j=1}^{T_k} \left\| H_k^{-1} \left( \varepsilon_{k,j}^{(m)} - \widetilde{\varepsilon}_{k,j}^{(m)} \right) \right\|_2^2 \right\}^{1/2}$$

$$\lesssim \left( \sum_{j=1}^{T_k} \eta_{k,j}^{1/2} \right)^{1/2} \lesssim T_k^{\{1-a/2\}/2}.$$

Hence, for any $1 \leq m \leq M_k$,

$$\left\| \mathcal{T}_{k,4}^{(m)} \right\|_2 = \left\| \frac{1}{T_k} \sum_{j=1}^{T_k} H_k^{-1} \left( \varepsilon_{k,j}^{(m)} - \widetilde{\varepsilon}_{k,j}^{(m)} \right) \right\|_2 = \mathcal{O}\left( T_k^{-1/2-a/4} \right).$$

**For $\mathcal{T}_{k,5}$:** Applying Theorem 2.6.7 of Csörgo & Révész (1981) with $H(x) = x^{2p}$ and $x_n = vT_k^{\beta_0}$, there exist i.i.d. standard normal $\widetilde{Z}_{k,j}^{(m)}$'s and some $a_k, C_k > 0$ (depending on the distribution of $H_k^{-1} \widetilde{\varepsilon}_{k,j}^{(m)}$) such that

$$\mathbb{P} \left( \left| \sum_{j=1}^{T_k} \frac{H_k^{-1} \widetilde{\varepsilon}_{k,j}^{(m)}}{\sqrt{H_k^{-1} S_k H_k^{-1}}} - \sum_{j=1}^{T_k} \widetilde{Z}_{k,j}^{(m)} \right| > vT_k^{\beta_0} \right) \leq C_k a_k^{-2p} v^{-2p} T_k^{1-2p\beta_0}.$$

Thus,

$$\mathbb{P}\left(\left|\frac{1}{T_k}\sum_{j=1}^{T_k}\frac{H_k^{-1}\widetilde{\varepsilon}_{k,j}^{(m)}}{\sqrt{H_k^{-1}S_kH_k^{-1}}} - \frac{1}{T_k}\sum_{j=1}^{T_k}\widetilde{Z}_{k,j}^{(m)}\right| > vT_k^{-1+\beta_0}\right) \lesssim v^{-2p}T_k^{1-2p\beta_0}.$$

Since $\mathbb{E}X^p = \int_0^\infty pv^{p-1}\mathbb{P}(|X| > v)dv$, we also have

$$\left\|\mathcal{T}_{k,5}^{(m)} - \frac{1}{T_k}\sum_{j=1}^{T_k}Z_{k,j}^{(m)}\right\|_2 = \mathcal{O}\left(T_k^{-1+\beta_0}\right).$$

Acoording to the above results, we show that

$$\left\|\mathcal{T}_k^{(m)} - \frac{1}{T_k}\sum_{j=1}^{T_k}Z_{k,j}\right\|_2 = o\left(T_k^{-1/2}\right),$$

which implies

$$\left\|\mathcal{T}_k^{(m)}\right\|_2 = \left\|\frac{1}{\sqrt{T_k}}\sum_{j=1}^{T_k}Z_{k,j}\right\|_2 + o(1) < \infty.$$

The SLLN (i.i.d.) and the continuous mapping theorem further yields that

$$\frac{1}{M_k}\sum_{m=1}^{M_k}\left(\sqrt{T_k}\mathcal{T}_k^{(m)}\right)^2 \xrightarrow{a.s.} \sigma_k^2, \quad \left(\frac{1}{M_k}\sum_{m=1}^{M_k}\sqrt{T_k}\mathcal{T}_k^{(m)}\right)^2 \xrightarrow{a.s.} 0,$$

which completes the proof of consistency of $\widehat{\sigma}_k^2$.

## D.2 PROOF OF LEMMA C.4

Under Assumption A.4, it is easy to obtain that for every $0 \le k \le K$,

$$|\widehat{b}_k - b_k| = \mathcal{O}_p\left(\frac{1}{\sqrt{N}}\right); \quad |\widehat{b}_k^2 - b_k^2| = \mathcal{O}_p\left(\frac{1}{N}\right) + |b_k|\mathcal{O}_p\left(\frac{1}{\sqrt{N}}\right). \tag{D.1}$$

Next, define the notation $\lesssim_p$ as follows: $a \lesssim_p b$ indicates that $a \lesssim b$ holds with probability $1 - o(1)$. Recall the definitions of $w_k^*$ and $\widehat{w}_k$. Define $\Delta_k = \frac{\sigma_k^2}{n_k} + \lambda b_k^2$, and $\widehat{\Delta}_k = \frac{\widehat{\sigma}_k^2}{n_k} + \lambda\widehat{b}_k^2$. Then, we have

$$w_k^* = \frac{\Delta_k^{-1}}{\sum_{j=0}^K\Delta_j^{-1}}, \quad \widehat{w}_k = \frac{\widehat{\Delta}_k^{-1}}{\sum_{j=0}^K\widehat{\Delta}_j^{-1}}.$$

We first consider the case $K = 2$ and scenario (c). i.e., $\lambda$ satisfies Assumption 1, and the bias scale satisfies Assumption 2. In this case, $w_k^*$ and $\widehat{w}_k$ can be rewritten as

$$w_k^* = \frac{\Delta_{k_1}\Delta_{k_2}}{\Delta_0\Delta_1 + \Delta_0\Delta_2 + \Delta_1\Delta_2} = \frac{\Delta_{k_1}\Delta_{k_2}}{\sum_{k=0}^2\prod_{j\neq k}\Delta_j}, \widehat{w}_k = \frac{\widehat{\Delta}_{k_1}\widehat{\Delta}_{k_2}}{\sum_{k=0}^2\prod_{j\neq k}\widehat{\Delta}_j} \text{ for } k_1 \neq k_2, k_1 \neq k, k_2 \neq k.$$

It can then be verified that

$$
\begin{aligned}
|\widehat{w}_k - w_k^*| &= \left|\frac{\Delta_{k_1}\Delta_{k_2}}{\sum_{k=0}^2\prod_{j\neq k}\Delta_j} - \frac{\widehat{\Delta}_{k_1}\widehat{\Delta}_{k_2}}{\sum_{k=0}^2\prod_{j\neq k}\widehat{\Delta}_j}\right| \\
&= \frac{\left|(\widehat{\Delta}_{k_1}\widehat{\Delta}_{k_2} - \Delta_{k_1}\Delta_{k_2})(\sum_{k=0}^2\prod_{j\neq k}\Delta_j) + \Delta_{k_1}\Delta_{k_2}\left\{\sum_{k=0}^2(\prod_{j\neq k}\Delta_j - \prod_{j\neq k}\widehat{\Delta}_j)\right\}\right|}{(\sum_{k=0}^2\prod_{j\neq k}\Delta_j)(\sum_{k=0}^2\prod_{j\neq k}\widehat{\Delta}_j)} \\
&\le \frac{|\widehat{\Delta}_{k_1}\widehat{\Delta}_{k_2} - \Delta_{k_1}\Delta_{k_2}| + |\sum_{k=0}^2(\prod_{j\neq k}\Delta_j - \prod_{j\neq k}\widehat{\Delta}_j)|}{\sum_{k=0}^2\prod_{j\neq k}\Delta_j}\frac{\sum_{k=0}^2\prod_{j\neq k}\Delta_j}{\sum_{k=0}^2\prod_{j\neq k}\widehat{\Delta}_j}.
\end{aligned}
$$

The inequality holds because $\Delta_{k_1}\Delta_{k_2} \leq \sum_{k=0}^2 \prod_{j \neq k} \Delta_j$. Under assumption A.4, we know that $\widehat{\Delta}_j$ is a consistent estimator of $\Delta_j$. Therefore, we obtain

$$|\widehat{w}_k - w_k^*| \lesssim_p \frac{|\widehat{\Delta}_{k_1}\widehat{\Delta}_{k_2} - \Delta_{k_1}\Delta_{k_2}| + |\sum_{k=0}^2(\prod_{j \neq k}\Delta_j - \prod_{j \neq k}\widehat{\Delta}_j)|}{\sum_{k=0}^2 \prod_{j \neq k}\Delta_j} \lesssim \frac{\sum_{k=0}^2 |\prod_{j \neq k}\widehat{\Delta}_j - \prod_{j \neq k}\Delta_j|}{\sum_{k=0}^2 \prod_{j \neq k}\Delta_j}.$$
(D.2)

We now analyze the numerator in equation (D.2). Under assumption A.4 and equation (D.1), we have

$$\left|\widehat{\Delta}_k - \Delta_k\right| \lesssim \frac{|\widehat{\sigma}_k^2 - \sigma_k^2|}{N} + \lambda|\widehat{b}_k^2 - b_k^2| \lesssim_p \frac{\mathcal{O}(1)}{N} + \frac{\lambda|b_k|}{\sqrt{N}}.$$
(D.3)

It can thus be shown that

$$\left|\widehat{\Delta}_{k_1}\widehat{\Delta}_{k_2} - \Delta_{k_1}\Delta_{k_2}\right| \leq \left|\Delta_{k_1}(\Delta_{k_2} - \widehat{\Delta}_{k_2})\right| + \left|\Delta_{k_2}(\Delta_{k_1} - \widehat{\Delta}_{k_1})\right| + \left|(\widehat{\Delta}_{k_2} - \Delta_{k_2})(\Delta_{k_1} - \widehat{\Delta}_{k_1})\right|$$

$$\lesssim_p \left(\frac{\sigma_{k_1}^2}{N} + \lambda b_{k_1}^2\right)\left\{\frac{\lambda|b_{k_2}|}{\sqrt{N}} + \frac{\mathcal{O}(1)}{N}\right\} + \left(\frac{\sigma_{k_2}^2}{N} + \lambda b_{k_2}^2\right)\left\{\frac{\lambda|b_{k_1}|}{\sqrt{N}} + \frac{\mathcal{O}(1)}{N}\right\} + \left\{\frac{\lambda|b_{k_1}|}{\sqrt{N}} + \frac{\mathcal{O}(1)}{N}\right\} \times$$

$$\left\{\frac{\lambda|b_{k_2}|}{\sqrt{N}} + \frac{\mathcal{O}(1)}{N}\right\} = \mathcal{E}_1 + \mathcal{E}_2 + \mathcal{E}_3.$$

If $|b_{k_1}| \ll N^{-1/2}$, we have

$$\frac{\lambda|b_{k_1}|}{\sqrt{N}} + \frac{\mathcal{O}(1)}{N} \ll \frac{\sigma_{k_1}^2}{N} + \lambda b_{k_1}^2,$$

which implies $\mathcal{E}_3 \ll \mathcal{E}_1$. Similarly, if $|b_{k_2}| \ll N^{-1/2}$, we have $\mathcal{E}_3 \ll \mathcal{E}_2$. If both $\lambda|b_{k_1}| \gg N^{-1/2}$ and $\lambda|b_{k_2}| \gg N^{-1/2}$, then

$$\frac{\mathcal{E}_3}{\mathcal{E}_1} = \mathcal{O}\left(\frac{\lambda^2|b_{k_1}||b_{k_2}|}{N} \times \frac{\sqrt{N}}{\lambda b_{k_1}^2 \lambda|b_{k_2}|}\right) = \mathcal{O}\left(\frac{1}{\sqrt{N}|b_{k_1}|}\right) = o(1).$$

Thus, we simplify to obtain

$$\left|\widehat{\Delta}_{k_1}\widehat{\Delta}_{k_2} - \Delta_{k_1}\Delta_{k_2}\right| \lesssim_p \left|\Delta_{k_1}(\widehat{\Delta}_{k_2} - \Delta_{k_2})\right| + \left|\Delta_{k_2}(\widehat{\Delta}_{k_1} - \Delta_{k_1})\right|$$

$$\lesssim_p \left(\frac{\sigma_{k_1}^2}{N} + \lambda b_{k_1}^2\right)\left\{\frac{\lambda|b_{k_2}|}{\sqrt{N}} + \frac{\mathcal{O}(1)}{N}\right\} + \left(\frac{\sigma_{k_2}^2}{N} + \lambda b_{k_2}^2\right)\left\{\frac{\lambda|b_{k_1}|}{\sqrt{N}} + \frac{\mathcal{O}(1)}{N}\right\}.$$

Next, we analyze the denominator in equation (D.2). By definition, it could be shown that

$$\sum_{k=0}^2 \prod_{j \neq k} \Delta_j \asymp \sum_{k=0}^2 \prod_{j \neq k}\left(\frac{\sigma_j^2}{N} + \lambda b_j^2\right).$$

Substituting this result back into equation (D.2), we obtain

$$|\widehat{w}_k - w_k^*| \lesssim_p \left\{\sum_{k=0}^2 \prod_{j \neq k}\left(\frac{\sigma_j^2}{N} + \lambda b_j^2\right)\right\}^{-1}\left\{\sum_{k=0}^2 \sum_{i \neq k}\left(\frac{\lambda|b_i|}{\sqrt{N}} + \frac{o(1)}{N}\right)\prod_{j \neq i}^{j \neq k}\left(\frac{\sigma_j^2}{N} + \lambda b_j^2\right)\right\}.$$

We now consider different cases. First, note that we have $b_0 = 0$. For $k = 1, 2$, we discuss the following scenarios: (1) When $b_k \ll N^{-1/2}$ for $k = 1, 2$, the denominator is of order $1/N^2$, while the numerator is of order $o(1)/N^2$. (2) When $\lambda b_k \gg N^{-1/2}$ for one particular $k$ and $b_j \ll N^{-1/2}$ for $j \neq k$, the denominator has order $\frac{\lambda b_k^2}{N}$, while the numerator has order $(\lambda b_k^2)o(1)/N + (\lambda|b_k|)/(N^{3/2})$. (3) When $\lambda b_k \gg N^{-1/2}$ for both $k = 1, 2$, the denominator has order $(\lambda b_1^2)(\lambda b_2^2)$, while the numerator is of order $(\lambda b_1^2)(\lambda|b_2|/\sqrt{N}) + (\lambda b_2^2)(\lambda|b_1|/\sqrt{N})$. In each scenario above, we can verify that $|\widehat{w}_k - w_k^*| = o_p(1)$.

For the general case with arbitrary $K$, the proof follows analogously. The detailed argument is omitted here, but it similarly leads to the result:

$$|\widehat{w}_k - w_k^*| \lesssim_p \left\{\sum_{k=0}^K \prod_{j \neq k}\left(\frac{\sigma_j^2}{N} + \lambda b_j^2\right)\right\}^{-1}\left\{\sum_{k=0}^K \sum_{i \neq k}\left(\frac{\lambda|b_i|}{\sqrt{N}} + \frac{o(1)}{N}\right)\prod_{j \neq i}^{j \neq k}\left(\frac{\sigma_j^2}{N} + \lambda b_j^2\right)\right\}.$$

Under Assumption 2, we therefore conclude that $|\widehat{w}_k - w_k^*| = o_p(1)$. Finally, noting that scenarios (a) and (b) are in fact special cases of scenario (c), we complete the proof of the theorem.

### D.3 PROOF OF THEOREM 4.2

The proof of the consistency of $\widehat{w}_k$ is provided in Appendix D.2. Next, we establish the asymptotic properties of the proposed estimator $\widehat{\theta}_{\text{est}}$. The proof is divided into four parts. In the first part, we prove that:

$$\sqrt{N}(\widehat{\theta}(\mathbf{w}^*) - \theta_0) \to_d \mathcal{N}\big(0, \sigma^2(\mathbf{w}^*)\big).$$

In the second part, we establish that:

$$\sqrt{N}(\widehat{\theta}(\mathbf{w}^*) - \widehat{\theta}_{\text{est}}) = o_p(1).$$

Combining the results from the first two parts immediately yields:

$$\sqrt{N}(\widehat{\theta}_{\text{est}} - \theta_0) \to_d \mathcal{N}\big(0, \sigma^2(\mathbf{w}^*)\big).$$

In the third part, we prove:

$$\widehat{\sigma}^2_{\text{est}} \to_p \sigma^2(\mathbf{w}^*).$$

Last, we show that data integration can improve efficiency. By combining these four parts, we directly obtain Theorem 4.2.

PART 1. Recall that the estimator $\widehat{\theta}(\mathbf{w})$ can be written as:

$$\widehat{\theta}(\mathbf{w}) = \sum_{k=0}^{K} w_k \widehat{\theta}_k = \widehat{\theta}_0 + \sum_{k=1}^{K} w_k \widehat{b}_k,$$

where $\widehat{b}_k = \widehat{\theta}_k - \widehat{\theta}_0$. Define the true weighted parameter $\theta^*(\mathbf{w}) = \sum_{k=0}^{K} w_k \theta_k = \theta_0 + \sum_{k=1}^{K} w_k b_k$. We first show that $\sqrt{N}(\theta^*(\mathbf{w}^*) - \theta_0) = o(1)$. To see this, consider two cases based on Assumption 2: Frist, for $b_k \ll N^{-1/2}$, it follows immediately that: $\sqrt{N} b_k = o(1)$. In addition, for $b_k \gg N^{-1/2}$, recall the oracle weight definition:

$$w_k^* = \frac{\big(\frac{\sigma_k^2}{n_k} + \lambda b_k^2\big)^{-1}}{\big(\frac{\sigma_0^2}{n_0}\big)^{-1} + \sum_{j=1}^{K}\big(\frac{\sigma_j^2}{n_j} + \lambda b_j^2\big)^{-1}}.$$

Under Assumption 1, we obtain: $w_k^* = O(1/(\lambda b_k^2 N)) = o(1)$, which implies:

$$\sqrt{N} w_k^* b_k = O\big(\sqrt{N} \frac{1}{\lambda b_k^2 N} b_k\big) = O\big(\frac{1}{\lambda |b_k| \sqrt{N}}\big) = o(1). \tag{D.4}$$

Therefore, under Assumption 2, we have:

$$\sqrt{N}(\theta^*(\mathbf{w}) - \theta_0) = o(1).$$

Combining this with Lemma C.2, we conclude that:

$$\sqrt{N}(\widehat{\theta}(\mathbf{w}^*) - \theta_0) \to_d \mathcal{N}(0, \sigma^2(\mathbf{w}^*)).$$

This finishes PART 1.

PART 2. Next, we analyze: $\sqrt{N}\big\{\widehat{\theta}(\mathbf{w}^*) - \widehat{\theta}_{\text{est}}\big\}$. By definition, we have

$$\widehat{\theta}(\mathbf{w}^*) - \widehat{\theta}_{\text{est}} = \widehat{\theta}_0 + \sum_{k=1}^{K} w_k^*(\widehat{\theta}_k - \widehat{\theta}_0) - \Big\{\widehat{\theta}_0 + \sum_{k=1}^{K} \widehat{w}_k(\widehat{\theta}_k - \widehat{\theta}_0)\Big\} = \sum_{k=1}^{K}(w_k^* - \widehat{w}_k)(\widehat{\theta}_k - \widehat{\theta}_0).$$

Again, we consider two scenarios: When $b_k \ll N^{-1/2}$, it can be verified from equation (C.1) that:

$$\sqrt{N}(\widehat{\theta}_k - \widehat{\theta}_0) = \sqrt{N}(\widehat{b}_k - b_k) + \sqrt{N} b_k = O_p(1).$$

Using Lemma C.4 and Assumption A.3, we thus have:

$$\sqrt{N}(w_k^* - \widehat{w}_k)(\widehat{\theta}_k - \widehat{\theta}_0) = o_p(1).$$

For the case $b_k \gg N^{-1/2}$, we similarly decompose:

$$\sqrt{N}(w_k^* - \widehat{w}_k)(\widehat{\theta}_k - \widehat{\theta}_0) = \sqrt{N}(w_k^* - \widehat{w}_k)(\widehat{\theta}_k - \theta_k) - \sqrt{N}(w_k^* - \widehat{w}_k)(\widehat{\theta}_0 - \theta_0) + \sqrt{N}(w_k^* - \widehat{w}_k) b_k.$$

By Lemmas C.2 and Lemma C.4, the first two terms are $o_p(1)$. Thus, it suffices to study $\sqrt{N}(w_k^* - \widehat{w}_k)b_k$. Note that

$$\sqrt{N}(w_k^* - \widehat{w}_k)b_k = \sqrt{N}w_k^* b_k - \sqrt{N}\widehat{w}_k(b_k - \widehat{b}_k) - \sqrt{N}\widehat{w}_k \widehat{b}_k.$$

First, from the analysis in PART 1, we already have $\sqrt{N}w_k^* b_k = o_p(1)$ by equation (D.4). Next, recall:

$$\widehat{w}_k = \frac{(\frac{\widehat{\sigma}_k^2}{n_k} + \lambda \widehat{b}_k^2)^{-1}}{\frac{\widehat{\sigma}_0^2}{n_0} + \sum_{j=1}^{K}(\frac{\widehat{\sigma}_j^2}{n_j} + \lambda \widehat{b}_j^2)^{-1}}.$$

Using equation (C.1) again, it can be shown that when $b_k \gg N^{-1/2}$, $1/(\widehat{b}_k \sqrt{N}) = o_p(1)$ when $b_k \gg N^{-1/2}$. As a result, it could be shown that

$$\widehat{w}_k = O_p\Big(\frac{1}{\lambda \widehat{b}_k^2 N}\Big)$$

since $\widehat{\sigma}_k^2 - \sigma_k^2 = o_p(1)$. This yields

$$\sqrt{N}\widehat{w}_k \widehat{b}_k = O_p\Big(\frac{1}{\lambda \widehat{b}_k \sqrt{N}}\Big) = o_p(1).$$

Hence, under Assumption 2, we obtain:

$$\sqrt{N}\Big(\widehat{\theta}(\mathbf{w}^*) - \widehat{\theta}_{\text{est}}\Big) = o_p(1),$$

which completes PART 2.

PART 3. Subsequently, from Assumption A.4, we have: $\widehat{\sigma}_k^2 - \sigma_k^2 = o_p(1)$. Combining this with Lemma C.4, we have: $\widehat{\sigma}_{\text{est}}^2 \to_p \sigma^2(\mathbf{w}^*)$, which completes PART 3.

PART 4. Recall that the loss function is defined as: $\mathcal{L}(\mathbf{w}) = \sum_{k=0}^{K} w_k^2 \frac{\sigma_k^2}{n_k} + \lambda \sum_{k=0}^{K} w_k^2 b_k^2$. The corresponding optimization problem is given by:

$$\mathbf{w}^* = \arg\min_{\mathbf{w}} \mathcal{L}(\mathbf{w}), \quad \text{subject to} \quad w_k \geq 0 \text{ for all } k, \quad \sum_{k=0}^{K} w_k = 1. \tag{D.5}$$

Clearly, $(1, 0, \ldots, 0)$ is a feasible solution to problem (D.5), with the corresponding loss equal to $\sigma_0^2/n_0$. As a consequence, to prove Theorem 3, it suffices to show that there exists a feasible solution $\mathbf{w}'$ satisfying $\mathcal{L}(\mathbf{w}') < \sigma_0^2/n_0$.

Suppose that the bias scale of the $k$th site satisfies $b_k \ll N^{-1/2}$. We define $\mathbf{w}' = (w_j')$ as follows:

$$w_0' = \frac{\sigma_k^2/n_k}{\sigma_0^2/n_0 + \sigma_k^2/n_k}, \quad w_k' = \frac{\sigma_0^2/n_0}{\sigma_0^2/n_0 + \sigma_k^2/n_k}, \quad w_j' = 0 \text{ for all } j \neq 0, k.$$

Clearly, $\mathbf{w}'$ is feasible, and it could be computed that:

$$\mathcal{L}(\mathbf{w}') = \frac{(\sigma_0^2/n_0)(\sigma_k^2/n_k)}{\sigma_0^2/n_0 + \sigma_k^2/n_k} + \lambda w_k'^2 b_k^2 = \frac{1}{\frac{1}{\sigma_0^2/n_0} + \frac{1}{\sigma_k^2/n_k}}(1 + o(1)).$$

This yields:

$$\frac{\mathcal{L}(\mathbf{w}')}{\sigma_0^2/n_0} = \frac{\sigma_k^2/n_k}{\sigma_0^2/n_0 + \sigma_k^2/n_k}(1 + o(1)) < 1.$$

This finishes the whole theorem proof.

### D.4 PROOF OF THEOREM 4.3.

Note that the conservative weight $\widetilde{w}_k$ does not necessarily converge to the oracle weight $w_k^*$. Denote $U_0 = 0$, $U_k = \mathcal{C}\sqrt{\sigma_k^2/n_k + \sigma_0^2/n_0}$ for $1 \leq k \leq K$, we introduce the auxiliary weights

$$\widetilde{w}_k^* = \frac{\left\{\frac{\sigma_k^2}{n_k} + \lambda(|b_k| + U_k)^2\right\}^{-1}}{\left(\frac{\sigma_0^2}{n_0}\right)^{-1} + \sum_j \left\{\frac{\sigma_j^2}{n_j} + \lambda(|b_j| + U_j)^2\right\}^{-1}} \text{ for } 0 \leq k \leq K.$$

Using these weights, we define the auxiliary estimator $\widehat{\theta}(\widetilde{\mathbf{w}}) = \sum_{k=0}^{K} \widetilde{w}_k \widehat{\theta}_k$. The proof has two steps. in the first step, We show that $|\widetilde{w}_k - \widetilde{w}_k^*| = o_p(1)$. In the second step, we establish Theorem 4.3 with the help of $\widetilde{w}_k^*$ and $\widehat{\theta}(\widetilde{\mathbf{w}}^*)$.

STEP 1. Recall the definition of $\widetilde{w}_k$, denote $\widehat{U}_0 = 0$, $\widehat{U}_k = \mathcal{C}\sqrt{\widehat{\sigma}_k^2/n_k + \widehat{\sigma}_0^2/n_0}$ for $1 \le k \le K$, we have

$$
\widetilde{w}_k = \frac{\left\{\frac{\widehat{\sigma}_k^2}{n_k} + \lambda(|\widehat{b}_k| + \widehat{U}_k)^2\right\}^{-1}}{\left(\frac{\widehat{\sigma}_0^2}{n_0}\right)^{-1} + \sum_j \left\{\frac{\widehat{\sigma}_j^2}{n_j} + \lambda(|\widehat{b}_j| + \widehat{U}_j)^2\right\}^{-1}} \quad \text{for } 0 \le k \le K.
$$

Then we have

$$
\left|\widetilde{b}_k^2 - (|b_k| + U_k)^2\right| = \left|(|\widehat{b}_k| + \widehat{U}_k)^2 - (|b_k| + U_k)^2\right| \tag{D.6}
$$
$$
\le \left\{(|\widehat{b}_k| + \widehat{U}_k) - (|b_k| + U_k)\right\}^2 + (|b_k| + U_k)\left\{(|\widehat{b}_k| + \widehat{U}_k) - (|b_k| + U_k)\right\}.
$$

It could be proved that

$$
|\widehat{U}_k - U_k|^2 \le \mathcal{C}^2\left|\left(\frac{\widehat{\sigma}_k^2}{n_k} + \frac{\widehat{\sigma}_0^2}{n_0}\right) - \left(\frac{\sigma_k^2}{n_k} + \frac{\sigma_0^2}{n_0}\right)\right| \le \mathcal{C}^2\left(\left|\frac{\widehat{\sigma}_k^2}{n_k} - \frac{\sigma_k^2}{n_k}\right| + \left|\frac{\widehat{\sigma}_0^2}{n_0} - \frac{\sigma_0^2}{n_0}\right|\right).
$$

This yields

$$
|\widehat{U}_k - U_k| \le \mathcal{C}\left(\left|\frac{\widehat{\sigma}_k^2}{n_k} - \frac{\sigma_k^2}{n_k}\right| + \left|\frac{\widehat{\sigma}_0^2}{n_0} - \frac{\sigma_0^2}{n_0}\right|\right)^{1/2}.
$$

Because we assume $\mathcal{C} \to \infty$ and $\mathcal{C}\sqrt{\widehat{\sigma}_k^2 - \sigma_k^2} = \mathcal{O}_p(1)$ for every $k = 0, \ldots, K$, then we have

$$
\left|(|\widehat{b}_k| + \widehat{U}_k) - (|b_k| + U_k)\right| \le \left|\left|\widehat{b}_k\right| - |b_k|\right| + \left|\widehat{U}_k - U_k\right| \le \left|\widehat{b}_k - b_k\right| + \left|\widehat{U}_k - U_k\right| = \mathcal{O}_p\left(\frac{1}{\sqrt{N}}\right).
$$

Substituting the bound above into equation (D.6) yields

$$
\left|(|\widehat{b}_k| + \widehat{U}_k)^2 - (|b_k| + U_k)^2\right| = \mathcal{O}_p\left(\frac{1}{N}\right) + \left\{|b_k| + \mathcal{C}\Omega_p\left(\frac{1}{\sqrt{N}}\right)\right\}\mathcal{O}_p\left(\frac{1}{\sqrt{N}}\right).
$$

Next, by repeating the key steps in the proof of Lemma C.4, we can readily show that $\widetilde{w}_k - \widetilde{w}_k^* = \mathcal{O}_p(1)$. The detailed proof is therefore omitted.

STEP 2. Note that $|b_k| + U_k \gg N^{-1/2}$, then it could be verified that

$$
\widetilde{w}_k^* \lesssim \frac{\lambda^{-1}\left\{(|b_k| + U_k)^2\right\}^{-1}}{N + \sum_j \left\{\lambda(|b_j| + U_j)^2\right\}^{-1}} = \mathcal{O}\left(\frac{1}{\lambda(|b_k| + U_k)^2 N}\right) = o(1),
$$

and

$$
\sqrt{N}\widetilde{w}_k^* b_k = \mathcal{O}\left(\frac{b_k}{\lambda(|b_k| + U_k)^2\sqrt{N}}\right) = o(1).
$$

Similarly, we can prove that

$$
\widetilde{w}_k = O_p\left(\frac{1}{\lambda(|\widehat{b}_k| + \widehat{U}_k)^2 N}\right)
$$

and

$$
\sqrt{N}\widetilde{w}_k \widehat{b}_k = O_p\left(\frac{\widehat{b}_k}{\lambda(|\widehat{b}_k| + \widehat{U}_k)^2\sqrt{N}}\right) = o_p(1).
$$

Next, by repating the key steps in the proof of Theorem 4.2, it could be verifeid that

$$
\sqrt{N}(\widehat{\theta}(\widetilde{\mathbf{w}}^*) - \theta_0) \to_d \mathcal{N}(0, \sigma^2(\widetilde{\mathbf{w}}^*)), \quad \sqrt{N}(\widehat{\theta}(\widetilde{\mathbf{w}}^*) - \widehat{\theta}_{\text{cons}}) = o_p(1).
$$

Finally, note that

$$
\widehat{\sigma}_{\text{cons}}^2 \to_p \sigma^2(\widetilde{\mathbf{w}}^*)
$$

by Theorem 4.1 and using the fact that $\widetilde{w}_k - \widetilde{w}_k^* = o_p(1)$. By combining these three results, we finished the whole proof.

