# OpenReview forum: "Privacy-Aware Data Integration for Enhanced Quantile Inference under Heterogeneity"
_ICLR.cc/2026/Conference — Submitted to ICLR 2026_

### Official Review · Reviewer_3avL · 2025-10-17

**Soundness:** 3
**Presentation:** 2
**Contribution:** 2
**Rating:** 4
**Confidence:** 3

**Summary:**

This paper studies quantile estimation and inference under heterogeneous datasets, where each dataset contains a different number of data points generated through a locally differentially private (LDP) mechanism applied to underlying true data. The goal is to estimate the true quantile for a target site by leveraging not only its own dataset but also information from other sites. To achieve this, the authors propose a weighted estimator that aggregates individual quantile estimates obtained by applying parallel SGD on each dataset. The weight vector is determined by minimizing a loss function that accouns for both the asymptotic variance and the heterogeneity of true quantiles. The authors rigorously establish theoretical guarantees of the proposed estimator, such as consistency and normality. Extensive numerical experiments further demonstrate the empirical advantages of the proposed method over existing approaches.

**Strengths:**

1. The authors develop a general and systematic framework for privacy-aware quantile estimation and inference via data integration
1. The analysis of statistical properties of the proposed quantile estimator is thorough and technically deep.
2. The numerical experiments convincingly demonstrate the improved performance.

**Weaknesses:**

1. The introduction section (including the motivating example in Figure 1) does not appear well aligned with the formal model and may confuse readers regarding the problem formulation and practical scope. The introduction gives the impression that the setting involves (i) a federated learning framework with data generated via a LDP mechanism and (ii) the use of only source sites' sensitive data to estimate the target site's quantile. However, the actual model is not a federated learning setup and the target site's data are also sensitive. Overall, I believe that the paper would benefit from a more consistent presentation of the problem setup in the introduction.
2. The current discussion focuses on the proposed estimator's statistical properties. Although such discussion and analyses are important, given the paper's emphasis on privacy, more efforts could be put into examining the impact of privacy constraints. For example, the presentation of several theorems (e.g. Theorem 4.1, 4.2, 4.3) could be improved by explicitly incorporating the randomness level $r_k$ of the LDP mechanisms.
3. The paper's technical contributions could become easier to digest for readers if more intuitive explanations can be provided.
4. The privacy guarantee in Proposition 4.1 requires further justification. While each local randomizer meets LDP, the interactive nature of SGD may need composition theory to properly track cumulative privacy loss. The authors refer to [1] to support the privacy claim; however, as far as I can tell, the cited work only establishes the LDP guarantee for their local randomizer, not for the overall algorithm. Therefore, a detailed proof or a proper reference is needed for the claimed LDP guarantees of both PSGD and Algorithm A.2.


[1] Yi Liu, Qirui Hu, Lei Ding, and Linglong Kong. Online local differential private quantile inference via self-normalization. In International Conference on Machine Learning, pp. 21698–21714. PMLR, 2023.

**Questions:**

1. Could the authors provide an intuitive explanation for why the estimator is designed to be a weighted one that incorporates quantile estimates from source sites rather than solely relying on the target site's? This design and the corresponding results appear to be counterintuitive to me. Since the data generating processes at different sites are assumed to follow different distributions (see line 116, $\mathcal{P}_k,\forall k$), it is unclear why using these potentially irrelevant source sites' data could be beneficial.
2. It might be helpful to explicitly write out $r_k$ in the statement of your main theoretical results, e.g., Theorem 4.1 - 4.4. In particular, for theorem 4.3, it would be interesting to see how the reduced variance is related to $\{r_k\}_{k=0}^N$? Establishing the relationship could help reveal how the level of randomness introduced by LDP mechanisms affects the estimator’s efficiency.
3. In the numerical experiments, can you also show empirical evidence of the claimed asymptotic normality? This would strengthen the connection between the theoretical results and empirical findings.
4. Regarding the setup of how private data is generated. Why did you choose to perturb the binary indicator function, rather than assuming that true data points are perturbed by a LDP mechanism and then shared with the target site? The latter situation seems more practical for your motivating example. Will this choice significantly affect your analysis? (I guess yes, as the binary indicator function in gradients would be more complex.)

---

> ### Author Response · Authors · 2025-11-24
>
> We sincerely appreciate the time and effort you've invested in reviewing our paper. Your feedback is invaluable to us, and we have addressed each of your concerns below. Following your advice, we have carefully revised the manuscript, with all modifications highlighted in blue. Should you have any further questions, please do not hesitate to ask.
>
> **For Weaknesses**
>
> ---
>
> 1. **The .... scope.**
>
>     **Response:** Thank you for your valuable suggestions. We have revised the introduction to better motivate our problem and to more clearly emphasize our contributions. We have also reorganized the pseudocode of the main algorithm in the main body and removed the original Figure 1. We hope that the revised manuscript presents our ideas more clearly.
>
> ---
>
> 2. **The ... properties.**
>
>     **Response:** Thank you for the important suggestion. Following your advice, we have revised Theorems 4.1–4.4 to explicitly incorporate the randomness level $r_k$ induced by the LDP mechanisms. We have also added a dedicated discussion of the resulting privacy accuracy trade-off. Please see the revised Section 4 for details.
>
> ---
>
> 3. **The ....provided.**
>
>     **Response:** For Theorem 4.2, we add the following sentences provides the intuition behind:
>     "Theorem 4.2 rigorously establishes the thriving asymptotic normality of our estimator $\widehat{\theta}\_{\text{est}}$ in the absence of unfavorable source sites whose biases are of order $N^{-1/2}$.
>     From Theorem 4.2, we find that larger values of $r_k$ or smaller values of $b_k$ lead to larger optimal weights $w_k^*$, which in turn yield smaller asymptotic variance.
>     However, larger $r_k$ also correspond to weaker privacy guarantees, as shown in Proposition 4.1. This reflects a fundamental and complex privacy accuracy trade-off inherent in the weighted estimator."
>
>     For Theorem 4.3, we have added the following intuitive discussion to clarify why the proposed estimator $\widehat{\theta}_{{\text{est}}}$ improves efficiency:
>
>     "Since the target estimator converges at rate $N^{-1/2}$, any source site bias of smaller order becomes negligible relative to the asymptotic variance. Leveraging information from these sites enlarges the effective  sample size and turns mild heterogeneity into a blessing. As a result, in such scenarios the proposed method improves efficiency and yields narrower confidence intervals than relying solely on the target data."
>
>     For Theorem 4.4, we have added the following intuitive explanation:
>
>     " Compared with Theorem 4.2, Theorem 4.4 imposes no additional bias scale restrictions, allowing source sites whose biases are of order $N^{-1/2}$. A closer inspection of the estimator $\widehat{\theta}_{\text{cons}}$ shows that, regardless of the magnitude of $b_k$, the adjusted term $\widetilde b_k$ dominates $N^{-1/2}$ due to $C \to \infty$. This construction effectively prevents source sites with bias   of order $N^{-1/2}$, which may be viewed as harmful sources,  from affecting the  transfer step, ensuring that the conservative method remains robust across more heterogeneous settings."
>
> ---
>
> 4. **The .... justification.**
>
>     **Response:** Following your constructive suggestions, we have discussed the LDP guarantees of our procedure:  "Since the Algorithm B.1 enjoys $(\epsilon_k,0)$-LDP with $\epsilon_k = \log\{(1 + r_k)/(1 - r_k)\}$, our Algorithms 1 and B.2 have  $(\max_{1\leq k\leq K} \epsilon_k,0)$-LDP guarantees as a direct consequence."
>
>     In the following, we provide a more detailed proof: Since Algorithm B.1 satisfies $(\epsilon_k,0)$-LDP with  $\epsilon_k = \log\{(1+r_k)/(1-r_k)\}$, each site's internal perturbation mechanism is $(\epsilon_k,0)$-LDP. Moreover, because the privacy mechanisms are applied independently across different sites without any interaction effect, the overall privacy guarantee is determined by the most stringent site-level bound. Therefore, our Algorithms 1 and B.2 satisfy $(\max_{1\le k\le K}\epsilon_k,0)$-LDP.

---

> > ### Author Response · Authors · 2025-11-24
> >
> > **For Questions**
> >
> > ---
> >
> > 1. **Could ... explanation?**
> >
> >     **Response:** Thank you for the insightful question. Our goal is to use information from the source sites to obtain better estimation and inference for the target parameter $\theta_0$. Therefore, we do not require the distributions across sites to be identical. As long as some source sites have parameters sufficiently close to the target parameter, these sites are informative. By integrating such informative source sites, we effectively increase the amount of useful data and therefore benefit even when other source sites may be irrelevant.
> >     Specifically, we estimate both the bias term $(\widehat{\theta}_k - \widehat{\theta}_0)$ and the variance $\widehat{\sigma}_k^2$ at each site. These quantities form the basis of our weighting strategy: sites with small drift and small variance receive larger weights, while sites with large drift or large variance receive very small weights. This allows us to leverage informative sites and down weight uninformative ones. As a result, as long as informative source sites are available, the aggregated estimator achieves smaller MSE, coverage probabilities closer to 95\%, and narrower confidence intervals than using only the target site data.
> >
> > ---
> >
> > 2. **It.... results.**
> >
> >     **Response:** First, thank you for the important suggestion. Per your kind advice, we now have revised Theorems 4.1 - 4.4 to explicitly incorporate the randomness level $r_k$ induced by the LDP mechanisms. We have also added a dedicated discussion on the resulting privacy accuracy trade-off. Please see the revised Section 4 for details.
> >
> >     In particular, for Theorem 4.3 (which has been merged into Theorem 4.2 in the revised manuscript), we note the following:  if at least one source site has a bias  smaller than the order $N^{-1/2}$, the weighted estimator achieves a strictly smaller asymptotic variance than $\widehat{\theta}\_0$. Hence, even if all the remaining  $r\_j$'s ($j\neq k$) shrink to zero, the convergence rate of $\widehat \theta\_{\operatorname{est}}$ is then at least $(n\_0 r\_0^2+n\_k r\_k^2)^{-1/2}$.  As a result,  the proposed method improves efficiency and yields narrower confidence intervals than relying solely on the target site data.
> >
> > ---
> >
> > 3. **In .... normality?**
> >
> >     **Response:** Thank you for the helpful suggestion. The empirical coverage probabilities (ECPs) reported in the paper already provide direct evidence of the claimed asymptotic normality. Under the asymptotic normality established in Theorems 3 and 5, the 95\% confidence intervals should attain coverage close to the nominal 95\% level. Across all numerical experiments under appropriate bias levels, the ECPs of the proposed methods remain close to (or above) 95\%, which serves as empirical verification of our theoretical result. We have also expanded the corresponding discussion in Section 5.2.
> >
> > ---
> >
> > 4. **Regarding .... analysis?**
> >
> >     **Response:** Thank you for your interesting question. This issue highlights a key distinction between non-linear statistics such as quantile estimation and linear statistics such as mean estimation. For mean estimation, it is relatively straightforward to design an LDP mechanism that preserves, or can recover, the true value. In contrast, for quantile estimation, standard LDP mechanisms distort the underlying distribution, and substantial additional effort is required to recover the true distribution (e.g., via deconvolution), which typically yields only logarithmic convergence rates; see [1]. Hence, only in very special cases can one achieve a parametric rate quantile estimator based on LDP perturbed raw data points.
> >
> >     To address this challenge, one of our contributions is extending the quantile LDP mechanism from the single-site setting (see [2]) to data integration across multiple sites. Perturbing binary indicator functions aligns well with the structure of the quantile loss and enjoys certain optimality properties, as discussed in [2]. Therefore, directly sharing LDP perturbed true data points is not suitable for the problems we consider.
> >
> > **Reference**
> >
> > [1] Fan, J. (1991). On the optimal rates of convergence for nonparametric deconvolution problems. Annals of Statistics, 1991.
> >
> > [2] Liu, Y., Hu, Q., Ding, L., & Kong, L. (2023). Online local differential private quantile inference via self-normalization. In ICML 2023.

---

### Official Review · Reviewer_keEz · 2025-10-23

**Soundness:** 3
**Presentation:** 3
**Contribution:** 2
**Rating:** 4
**Confidence:** 3

**Summary:**

The authors consider the problem of data integration/transfer learning for quantile estimation under local DP.  The authors recognized that in the Average Stochastic Gradient Descent algorithm, there is an update involving binary terms that lends itself well to incorporating local DP via a randomized response approach.  They apply this approach per dataset and then pool estimates together to produce a final estimate.  The pooling leverages a weighting scheme to downweight data that don't aid in the original estimation task.

**Strengths:**

Combines several active areas: quantile estimation, privacy, and transfer learning/data integration.  Extensive mathematical results including asymptotic results for the estimators.  Interesting weight scheme for combining estimates.

**Weaknesses:**

Overall, the paper seems strong mathematically (except a few small points), but the motivation for the paper seems a bit weak.  Data integration/transfer learning is very prominent with complex models, but not for a single quantile.  Furthermore, the role of LDP in the analysis seems like a bit of an afterthought as in the end it seems to increase your variance a bit, but the much bigger focus is on the bias between datasets.  However, LDP will inflate the uncertainty of estimates dramatically, which can be missed when treating the epsilon/r as fixed in the asymptotic analysis.

**Questions:**

- Minor, but 3.1 is a property, not a definition (i.e. that's not the definition of a quantile).
- Thm 4.1 and Assumption 4 don't make sense to me.  There are still sample sizes on the RHS.  What are the limits being taken with respect to?
- Early on they don't really state what the problem is.  What are they trying to estimate?  Especially if there is "drift".  Based on later discussions, it's \theta_0 based on the definition of the bias term.  But I don't see this in the problem statement.
- What role does r_k play in the asymptotic?  Accuracy of the estimates is a function of both r_k and the sample size.  I think in modern DP papers it isn't reasonable to ignore it's contribution by treating it as fixed.
- This problem seems weakly motivated (data integration to estimate a simple quantile), but maybe the LDP requirement can help motivate it since it will add a lot of uncertainty?
- Sample sizes are quite large in the empirical work.
- When does LDP add so much noise that you are better off just working with no privacy and the target data?  This is discussed in Thm 4.3, but there is no mention of the role of privacy.

---

> ### Author Response · Authors · 2025-11-24
>
> We sincerely appreciate the time and effort you've invested in reviewing our paper. Your feedback is invaluable to us, and we have addressed each of your concerns below. Following your advice, we have carefully revised the manuscript, with all modifications highlighted in blue. Should you have any further questions, please do not hesitate to ask.
>
> **For Weaknesses**
>
> ---
>
> 1. **Overall, the paper seems strong mathematically (except for a few small points), but the motivation for the paper seems a bit weak.**
>
>
>    **Response:** Please refer to Q5.
> ---
>
> 2. **Furthermore, the role of LDP in the analysis seems somewhat like an afterthought.**
>
>    **Response:**
>    We clarify that LDP is not an afterthought in our framework, but rather plays a central and structural role in both the methodology and the theory. First, LDP directly enters the weighting rule. Recall that the oracle weight is
>    $$w\_k^*=\Bigg\\{\sum\_{j=0}^K\left(\frac{\sigma\_j^2}{n_j}+\lambda b\_j^2\right)^{-1}\Bigg\\}^{-1}
>    \left(\frac{\sigma\_k^2}{n\_k}+\lambda b\_k^2\right)^{-1}.$$
>    This expression contains two components: a variance term $\sigma_k^2 / n_k$ and a bias term $b_k^2$. Because the variance $\sigma_k^2$ depends explicitly on the LDP noise level through the response rate $r_k$, the weighting mechanism naturally integrates both privacy induced uncertainty and site bias. For example, sites with stronger privacy constraints (i.e., smaller $r_k$) may receive smaller weights.
>
>    In addition, LDP drives the need for a new variance estimation method. In practice, computing the weights $w_k^*$ requires consistent variance estimators. The classical plug-in approach is infeasible because the raw data needed to estimate $f_k$s, which are not accessible under LDP. This limitation is precisely why we propose a PSGD based method that automatically estimates the variance $\sigma_k^2$ while operating entirely under LDP constraints. We now make it clear in the revision.
>
> ---
>
> 3. **LDP will inflate the uncertainty of estimates dramatically, which can be missed when treating the $\varepsilon/r$ as fixed in the asymptotic analysis.**
>
>     **Response:** Please refer to Q4 and Q7.
>
> **For Questions**
>
> ---
>
> 1. **Minor, .... definition.**
>
>     **Response:** We thank the reviewer for the remark. (3.1) states that
>     $$
>     \theta\_k = \mathop{\arg\min}\_{\theta}  \mathbb{E}\_{x \sim \mathcal{P}\_k}\\{\ell(x,\theta)\\},
>     $$
>     which is indeed a property satisfied by the population minimizer rather than a definition of the quantile itself. We have revised the wording accordingly to avoid potential confusion.
>
> ---
>
> 2. **Theorem .... me.**
>
>    **Response:**  We thank the reviewer for pointing out this issue. We have now corrected the statements in Theorem 4.1 and Assumption 4 to
>    $$
>    \sqrt{n_k}\big(\widehat{\theta}_k - \theta_k\big) \stackrel{d}{\longrightarrow} \mathcal{N}\big(0, \sigma_k^2 \big),
>    $$
>    with $\sigma\_k^2= \big\\{ 4r\_k^2 f\_k^2(\theta\_k) \big\\}^{-1} \big\\{1 - r\_k^2(2\tau - 1)^2\big\\},$ so that the asymptotic limit is taken with respect to $ n_k \to \infty $, and no sample-size terms remain on the right-hand side.
>
> ---
>
> 3. **Early .... is.**
>
>     **Response:** Thank you for the helpful comment. We clarify that our goal is to estimate the target parameter $\theta_0$, which is the parameter of primary interest at the target site. When "drift" is present, the source sites satisfy $\theta_k = \theta_0 + b_k$, so we estimate both the bias term $(\widehat{\theta}_k - \widehat{\theta}_0)$ and the variance $\widehat{\sigma}_k^2$ at each site. These quantities form the basis of our data integration strategy: sites with negligible or small drift (and small variance) receive larger weights, whereas sites with large drift or large variance receive very small weights. This weighting allows us to effectively leverage the informative sites while down weighting the uninformative ones. As a result, as long as informative source sites are available, the aggregated estimator achieves smaller MSE, coverage probabilities closer to 95\%, and narrower confidence intervals than using only the target data.

---

> > ### Author Response · Authors · 2025-11-24
> >
> > **For Questions**
> >
> > ---
> >
> > 4. **What role .... asymptotics?**
> >
> >     **Response:** Formally speaking, our algorithm enjoys $(\max_{1 \le k \le K} \epsilon_k, 0)$-LDP with $\epsilon_k = \log\{(1+r_k)/(1-r_k)\}$. Here, the response rate $r_k$ controls the level of privacy protection, with smaller values corresponding to stronger privacy guarantees. Recalling the oracle weight
> >     $$ w\_k^* = \Bigg\\{\sum\_{j=0}^K \left(\frac{\sigma\_j^2}{n\_j} + \lambda b\_j^2 \right)^{-1} \Bigg\\}^{-1} \left( \frac{\sigma\_k^2}{n\_k} + \lambda b\_k^2 \right)^{-1},
> >     $$
> >     where
> >     $$ \sigma\_k^2 = \bigl\\{4 r\_k^2 f\_k^2(\theta\_k)\bigr\\}^{-1} \bigl\\{1 - r\_k^2(2\tau - 1)^2\bigr\\},
> >     $$
> >     and following your insightful suggestions, we have added a more detailed explanation of these theoretical insights.
> >
> >     "Theorem 4.2 rigorously establishes the asymptotic normality of $\widehat{\theta}\_{\text{est}}$ in the absence of unfavorable source sites whose biases are of order $N^{-1/2}$. It is shown that larger $r\_k$ or smaller $b\_k$ yield larger optimal weights $w\_k^*$ and hence a smaller asymptotic variance. Larger $r\_k$, however, correspond to weaker privacy guarantees (Proposition 4.1), reflecting the fundamental accuracy privacy trade-off inherent in the weighted estimator...." see the revised Section 4 for details.
> >
> > ---
> >
> > 5. **This problem ....quantile).**
> >
> >     **Response:** Thank you for the insightful comment. Our goal is not only to estimate a quantile but also to provide valid inference for it. Under LDP, a target-only estimator can suffer from substantial uncertainty as you correctly pointed out. Consequently, the resulting confidence intervals may be wide and may fail to achieve 95\% coverage. Data integration directly addresses this issue. By incorporating auxiliary sites, our ADP estimators yield much narrower confidence intervals and coverage probabilities much closer to 95%. For example, Figure B.6 shows that when the target site operates under a stringent privacy budget, the Target estimator’s ECP deviates significantly from 95% and the CI becomes wide, whereas our method produces valid and substantially tighter intervals. We have also revised the introduction to strengthen the motivation of the problem.
> >
> > ---
> >
> >
> > 6. **Sample .... work.**
> >
> >     **Response:** Thank you for raising this important point. We would like to clarify that the sample sizes used in our experiments are modest in the context of SGD based methods. It is common in the literature for stochastic optimization and quantile estimation to involve datasets on the order of tens of thousands to hundreds of thousands of observations (see, e.g., [1] [2] [3]). Moreover, the real dataset used in our empirical study contains over 200,000 records, which is fully comparable to the simulated settings. Thus, the sample sizes in our experiments are representative of practical applications and appropriate for evaluating the proposed approach.
> >
> > ---
> >
> > 7. **When .... data?**
> >
> >     **Response:** Thank you for the important comment. Theorem 4.3 (which has been merged into Theorem 4.2 in the revised manuscript) shows that if at least one source site $k$ has a bias  smaller than the order $N^{-1/2}$, the weighted estimator achieves a strictly smaller asymptotic variance than $\widehat{\theta}\_0$. Hence, even if all the remaining $r\_j$'s ($j\neq k$) shrink to zero, the convergence rate of $\widehat \theta\_{\operatorname{est}}$ is then at least $(n\_0 r\_0^2+n\_k r\_k^2)^{-1/2}$.  As a result,  the proposed method improves efficiency and yields narrower confidence intervals than relying solely on the target site data.
> >
> >     Your concern may arise in certain extreme cases. For example, if all source sites have $r_k = 0$, implying the strongest possible privacy protection, then $\sigma_k^2$ diverges and the oracle weights satisfy $w_k^* \approx 0$ for every source $k$, while $w_0^* \approx 1$. In this scenario, the performance of the weighted estimator is essentially the same as using only the target data with no privacy. We now have added detailed discussions below the theorems.
> >
> > **Reference**
> >
> > [1] Liu, Y., Hu, Q., Ding, L., & Kong, L. (2023). Online local differential private quantile inference via self-normalization. ICML 2023.
> >
> > [2] Zhu, W., Lou, Z., Wei, Z., & Wu, W. B. (2024). High confidence level inference is almost free using parallel stochastic optimization. arXiv:2401.09346.
> >
> > [3] Xie, C., Jin, K., Liang, J., & Zhang, Z. (2024). Asymptotic time-uniform inference for parameters in averaged stochastic approximation. arXiv:2410.15057.

---

### Official Review · Reviewer_VmFV · 2025-10-31

**Soundness:** 2
**Presentation:** 2
**Contribution:** 1
**Rating:** 4
**Confidence:** 3

**Summary:**

This paper introduces a privacy aware multi site framework for quantile estimation under local differential privacy and distributional shifts, building a variance and bias penalized weighted estimator with a conservative variant for borderline bias and estimating variances via parallel SGD chains that comply with local differential privacy. The method achieves consistency, asymptotic normality, and efficiency gains over single site estimation when at least one source is informative, and experiments on Normal and Cauchy simulations and a real wage dataset show improved coverage and shorter intervals, with limitations including reliance on multiple SGD chains and a fixed number of sites.

**Strengths:**

The paper tackles a timely and practically relevant problem of multi-site quantile estimation under local differential privacy, and provides an implementable weighting framework with parallel-SGD-based variance estimation. Theoretical results establish consistency, asymptotic normality, and efficiency gains over single-site estimation, and experiments on Normal and Cauchy simulations and a real wage dataset demonstrate improved coverage and shorter intervals. The presentation is clear, with explicit weight formulas and transparent privacy parameters, indicating strong practical impact for cross institutional analytics.

**Weaknesses:**

1. Novelty is weak; the method mainly recombines known ingredients including local differential privacy, weighted aggregation, and parallel SGD without a distinct new idea; please identify one concrete, verifiable contribution that prior distributed or LDP quantile methods do not achieve.

2. The privacy analysis is under-specified; please provide a clear epsilon and delta privacy guarantee.

3. Sensitivity is unreported; please report coverage and interval length as a function of the privacy budget.

4. The intermediate bias regime with bias ≈ N^{-1/2} may jeopardize validity; please add a finite-sample coverage bound for this case.

5. Experiments cover Gaussian and Cauchy simulations plus a single real-world dataset, with few strong privacy-preserving baselines; please add stronger baselines.

6. Theoretical guarantees assume a diverging number of PSGD chains and a fixed number of sites; please clarify behavior when the number of sites grows.

**Questions:**

see the detail in weaknesses.

---

> ### Author Response · Authors · 2025-11-24
>
> We sincerely appreciate the time and effort you've invested in reviewing our paper. Your feedback is invaluable to us, and we have addressed each of your concerns below. Following your advice, we have carefully revised the manuscript, with all modifications highlighted in blue. Should you have any further questions, please do not hesitate to ask.
>
> ### **Weaknesses.**
>
> 1. **Novelty .... achieve.**
>
>    **Response:**
>    Thank you for raising this important point. Our paper makes two concrete and verifiable contributions that, to the best of our knowledge, are not achieved by existing distributed or LDP quantile methods.
>
>    1. **Methodological contribution.**
>       Existing distributed approaches for quantile estimation either
>       (i) assume that the source-site shifts $b_k$ are vanishing or fully distinguishable, and therefore do not handle the intermediate regimes arising in heterogeneous federated systems; or
>       (ii) do not provide valid quantile inference under LDP constraints.
>       Our framework simultaneously addresses both issues by allowing arbitrary bias levels $b_k$ and delivering valid inference with consistent variance estimation.
>
>    2. **Theoretical contribution.**
>       A central technical challenge is that weighted aggregation requires estimating the variance of the quantile estimator. Unlike prior work that avoids this difficulty—e.g., [1] using a heavier limiting pivotal distribution or [2] relying on high-confidence inference—we derive a consistent variance estimator without splitting either the data or the privacy budgets. Consequently, we address both the restrictions imposed by LDP and the complications arising from the non-smooth quantile loss, establishing asymptotic normality of the final estimator together with a consistent variance estimator.
>
>       Following your suggestions, we have revised the introduction to more clearly emphasize our contributions.
>
> ---
>
> 2. **The privacy .... guarantee.**
>
>    **Response:**
>    A direct calculation shows that the algorithm enjoys  $(\max_{1\leq k\leq K}\epsilon_k,0)$-LDP with $\epsilon_k=\log\{(1+r_k)/(1-r_k)\}$.
>
> ---
>
> 3. **Sensitivity .... budget.**
>
>    **Response:**
>    Thank you for the suggestion. We now include a sensitivity analysis with respect to the privacy budget, reporting both coverage and interval length as functions of the truthful response rate. The results are provided in Appendix B.2.
>
> ---
>
> 4. **The intermediate .... case.**
>
>    **Response:**
>    Thank you for raising this important point.
>
>    *Empirically*, we have conducted a series of experiments including the intermediate-bias regime; see Section 5.2. The ADP(cons) estimator remains stable and achieves empirical coverage close to the nominal 95%, demonstrating robust finite-sample performance.
>
>    *Theoretically*, this work focuses on statistical inference under an asymptotic framework. Because our method relies on the variance estimator derived from PSGD, the current theory does not extend to finite-sample coverage guarantees for non-smooth quantile estimators under LDP in the regime $b_k \approx N^{-1/2}$. We agree that this is an important direction for future research and have added corresponding discussion in Section 6.
>
> ---
>
> 5. **Experiments .... baselines.**
>
>    **Response:**
>    Thank you for the helpful suggestion. We have added a stronger privacy-preserving baseline: **DPSGD** [1], which injects noise directly into stochastic gradients rather than using randomized response, providing a more competitive comparison; see Section 5.1.
>
>    In addition, we now include experiments under *stronger* privacy constraints, including the setting $r_k = 0.25$ for all sites, corresponding to \((0.51, 0)\)-LDP. These results are reported in Appendix B.2.
>
> ---
>
> 6. **Theoretical .... grows.**
>
>    **Response:**
>    Thank you for your constructive and insightful suggestions.
>
>    The current framework assumes a fixed number of sites $K$; extending the methodology to diverging $K$ is an interesting open problem. In particular, the strong Gaussian approximation used in Theorem 4.1 for a fixed $k$ can be strengthened to hold uniformly for all $1 \le k \le K_n$, provided that $K_n$ does not grow too quickly with $n$. Nevertheless, a gap remains in establishing the asymptotic normality of the estimators in Theorems 4.2–4.3 when the number of sites diverges.
>
>    We have added the corresponding discussion in the concluding remarks.
>
>
> [1] Shuang Song, Kamalika Chaudhuri, and Anand D Sarwate. Stochastic gradient descent with differentially private updates. In 2013 IEEE global conference on signal and information processing, pp. 245–248. IEEE, 2013.

---

### Official Review · Reviewer_bjkU · 2025-11-03

**Soundness:** 3
**Presentation:** 2
**Contribution:** 2
**Rating:** 4
**Confidence:** 3

**Summary:**

This paper studies locally differential private (LDP) quantile estimation in the presence of auxiliary data sources that may have different privacy requirements and different data distributions. It provides a method for LDP quantile estimation at each data source as well as a method for aggregating these estimates to augment that of the target data source. In addition to a (straightforward) privacy analysis, it proves several asymptotic utility guarantees about its algorithm and presents experiments on synthetic and real data.

**Strengths:**

The problem setup is plausible, and I didn't find existing work on it. The proposed approach is reasonable and, to my reading, doesn't rely on any particularly strong assumptions, and is an algorithm I can actually imagine being run in the real world. This is not so common in DP.

**Weaknesses:**

*Clarity*: One weakness of this paper is that it lacks a concise explanation of its algorithm in the main body, so a reader has to piece the algorithm together over the course of many paragraphs of text. I suggest that the authors think about a clearly signposted, coherent, and self-contained presentation of the algorithm that can go in the main body -- a diagram, pseudocode, a high-level sketch, something. This would be a better use of space than Figure 1 or 2, IMO. As is, the paper tries to convey a narrative about the algorithm's construction as a sequence of ideas, which is maybe how the research project was developed, but is IMO not the clearest way to describe the finished product.

*Experiments*: I didn't get much info from the experiments. I suggest dropping ADP(0) ($\lambda = 0$ seems to be a qualitatively different algorithm than what the paper is interested in) and ADP(cv) (cross-validation, unless accounted for in the DP guarantee, feels unrealistic for this kind of application) and focusing on ADP(1) and maybe ADP(cons). This would also make the plots in Figure 3 easier to parse. Also, unless I missed it, the final LDP guarantee is not actually specified in the plots, though at response rate 0.5 I think it ends up being a reasonable value. The experiments also seem to be missing a suggested interpretation -- the paper just describes the experiment setup, provides a plot, and moves on to the conclusion. It looks like there might be a 1-2 order of magnitude improvement in the MSE, but I'm not sure how useful that is -- are we just making an already good enough estimate even better?

I would also suggest moving the real dataset experiments into the main body and providing a similar discussion, though I was confused by its presentation in Table A.3 -- are any of the provided metrics actually error? It looks like the experiment just records the values returned by different approaches, but says nothing about how good they are.

*Incremental?*: My understanding is that the overall algorithm is to run several instances of locally randomized ASGD on disjoint partitions of the data at each site (which it calls PSGD), use the end results to obtain quantile and variance estimates, and then aggregate these estimates by weighting according to bias and variance. As the paper notes, locally randomized ASGD (or a single data source has previously appeared in Liu et al. 2023, and PSGD appeared in Zhu-Li-Wang 2024 (and is anyway an instance of the classic subsample-and-aggregate algorithm). The paper's contribution is the weighted aggregation, which is logical but not so novel.

**Questions:**

See "Weaknesses". Overall, I think the paper makes a decent but limited contribution that is muddied by unclear presentation. A significantly revised version of the paper could be acceptable at ICLR or a similar venue, but I can't champion the current version, and I think even a significantly polished version would be a ~weak accept.

---

> ### Author Response · Authors · 2025-11-24
>
> We sincerely appreciate the time and effort you've invested in reviewing our paper. Your feedback is invaluable to us, and we have addressed each of your concerns below. Following your advice, we have carefully revised the manuscript, with all modifications highlighted in blue. Should you have any further questions, please do not hesitate to ask.
>
> ### **Weaknesses**
>
> **Clarity:**
> One weakness... product.
>
> **Response:**
> Thank you for your valuable suggestions. We first revised the introduction to better motivated our problem and emphasize the contribution. Then we reorganize the pseudocode of main algorithm (see Algorithm 1) in main body and remove the original Figure 1. We hope the revised manuscript can better present our idea.
>
>
>
> **Experiments:**
> I did not ....they are.
>
> **Response:**
> We apologize for the confusion in the experimental section. We thank the reviewer for the helpful suggestions and have revised the paper accordingly.
>
> 1. We agree that ADP(cv) is not appropriate in the LDP setting. We have removed ADP(cv) and replaced it with ADP(decay), using $\lambda = 8 \log N / \sqrt{N}$, which satisfies Assumption 1 when the shifts $b_k$ are of constant order; see the revised Section 5.2. ADP(0) is now treated only as a reference benchmark, rather than a method within our framework. We hope this change makes Figures clearer and easier to interpret.
>
> 2. We now explicitly report the LDP level at the beginning of the experimental setting. For example, $r = 0.5$ corresponds to $(1.10, 0)$-LDP.
>
> 3. We have added detailed explanations for all experiments in the revised Section 5.2. Each figure is now accompanied by a clear interpretation of the results and their implications.
>
> 4. Our goal is not only to reduce MSE but also to improve statistical inference. The proposed estimators yield narrower confidence intervals and coverage probabilities much closer to 95%, especially when the target site requires strong privacy. In such cases, the Target estimator alone provides poor inference. For example, Figure B.6 shows that when the target site operates under a stringent privacy budget, the Target estimator’s ECP deviates significantly from 95% and the CI becomes wide, whereas our method produces valid and substantially tighter intervals.
>
> 5. Because the real dataset has no ground truth, we cannot report MSE or coverage. Instead, we present the estimates together with their confidence interval lengths. We observe that CI lengths decrease as privacy is relaxed, and the intervals from different methods substantially overlap, suggesting stable behavior and practical usefulness. We have also moved the real data experiment into the main paper and added a clear discussion.
>
> 6. The revised Section 5.2 now provides a clearer interpretation of the experimental findings and highlights the advantages of the proposed methods.
>
> ---
>
> **Incremental:**
> My understanding .... novel.
>
> **Response:**
> Thank you for your comment. In this paper, our goal is to develop an enhanced inference procedure under the LDP framework via data integration. Beyond establishing the necessary inference theory, a key challenge is that weight aggregation requires estimating the variance of the quantile estimator. Unlike prior work that avoids this issue, e.g., [1] relying on a heavier limiting pivotal distribution or [2] using high-confidence inference, we derive a consistent variance estimator without splitting either data or privacy budgets. Consequently, we address both the constraints imposed by LDP and the difficulties arising from the non-smooth quantile loss, establishing asymptotic normality of the final estimator along with a consistent variance estimator. This is not a direct application of existing methods, but instead requires nontrivial new derivations. According to your suggestions, we have revised the introduction to more clearly emphasize our contributions.
>
> ---
>
> ### **Questions**
>
> See “Weaknesses.”.....
>
> **Response:**
>
> Thank you very much for your  constructive feedback.  Following your suggestions, we have substantially revised the manuscript to improve its organization and more clearly highlight the contributions and methodological innovations. We hope that these revisions address the concerns raised and lead to a clearer and more compelling presentation of our results.
>
> [1] Yi Liu, Qirui Hu, Lei Ding, and Linglong Kong. Online local differential private quantile inference
> via self-normalization. In International Conference on Machine Learning, pp. 21698–21714.
> PMLR, 2023.
>
> [2] Wanrong Zhu, Zhipeng Lou, Ziyang Wei, and Wei Biao Wu. High confidence level inference is almost free using parallel stochastic optimization. arXiv preprint arXiv:2401.09346, 2024.

---

### Author Response · Authors · 2025-12-02

We thank the reviewers for their time and insightful feedback on our paper, which, to the best of our knowledge, is the first to establish enhanced quantile inference  via data integration under heterogeneity and potential LDP.

Across the reviews, **there is broad recognition of the rigor and practical relevance of this work**.

- Reviewer bjkU finds the problem setup plausible, emphasizes that no existing work appears to address it, and notes that the proposed method relies on realistic assumptions and is implementable in real world DP settings, which is an uncommon strength.

- Reviewer VmFV highlights that the paper tackles a timely and practically important problem of multi-site quantile estimation under LDP, praises the implementable weighting and parallel SGD variance estimation framework, and commends the theoretical guarantees of consistency, asymptotic normality, and efficiency gains, as well as the strong empirical performance on both synthetic and real datasets.

- Reviewer 3avL similarly underscores the general and systematic nature of the proposed privacy-aware framework, the technically deep theoretical analysis, and the convincing numerical improvements.

- Reviewer keEz remarks that the mathematical development is strong, and although they raise concerns about motivation and the role of LDP, they acknowledge the solidity of the theoretical work and the attention to dataset heterogeneity.

Collectively, the reviewers recognize the submission as a mathematically rigorous and practically implementable contribution to privacy-aware federated quantile estimation.

---

The reviewers also raised valuable concerns regarding novelty, clarity of presentation, explanations of theoretical results, and numerical evaluations. We have thoroughly addressed these in our rebuttal. **A summary of the revisions is as follows.**

**Presentation and discussion:**

We substantially revised the introduction to better motivate the problem and to highlight our contributions and novelty. We also revised the theorems to explicitly incorporate the response rate $r_k$ induced by the LDP mechanisms, thereby clarifying the relationship between the theory and the privacy guarantees. In addition, we enhanced the theoretical explanations and intuitions and improved the overall presentation for better readability.

**Numerical experiments:**

We have reorganized the comparison setup in the simulation studies. In addition, we added a stronger privacy preserving baseline, incorporated experiments under stronger privacy constraints, and included a sensitivity analysis with respect to the privacy budget.

We have carefully revised the manuscript to incorporate the corresponding improvements, with all modifications highlighted in blue. We sincerely appreciate the constructive feedback and believe that these revisions significantly strengthen the paper.

---

### Meta-Review · Area_Chair_PCxQ · 2025-12-16

**Summary:**

1. Reviewer bjkU (Score: 4, Confidence: 3): The reviewer criticized the paper's novelty as a significant weakness. The reviewer wrote that even with a major modification to improve the presentation, the novelty is weak, and hence the reviewer would put "weak accept" at best.

2. Reviewer VmFV (Score: 4, Confidence: 3): The reviewer wrote "Novelty is weak".

3. Reviewer keEz (Score: 4, Confidence: 3): The reviewer pointed out that Thm 4.1 has a typo (appearing sample size "n"), which the authors corrected. Given that this is one of the main results of the paper, having typos in the important quantities seems sloppy. The reviewer also noted that the paper's motivation is weak. The authors responded to this point and updated the introduction accordingly.

4. Reviewer 3avL (Score: 4, Confidence: 3): The core comment I find is how the level of randomness introduced by LDP mechanisms affects the estimator’s efficiency. The authors incorporated this request and revised Theorems 4.1-4.4 to explicitly incorporate the randomness level introduced by the LDP mechanism.

Overall, while other contributions, such as updated Theorems and improved presentations, are worthwhile, the work is indeed weak in terms of novelty, as two of the reviewers mentioned. Hence, I recommend rejecting the paper.

**Reviewer Concerns:**

Reviewer 3avL (Score: 4, Confidence: 3): The core comment I find is how the level of randomness introduced by LDP mechanisms affects the estimator’s efficiency. The authors incorporated this request and revised 4.1-4.4 to explicitly incorporate the randomness level introduced by the LDP mechanism.

**Reviewer Scores:**

I am unsure whether the authors' responses to Reviewers bjkU and VmFV, who criticized the work for limited novelty, are satisfactory. If I were the reviewer, I would probably not change my score.

If I were Reviewer 3avL, I would probably increase the score to 6, thanking the revised Theorems 4.1-4.4 that incorporate the randomness level introduced by the LDP mechanism.

---

### Decision · Program_Chairs · 2026-01-26

Reject